# A Norrin/Wnt surrogate antibody stimulates endothelial cell barrier function and rescues retinopathy

Rony Chidiac[1,†] (iD), Md. Abedin[2,†], Graham Macleod[1] (iD), Andy Yang[1], Pierre E Thibeault[1], Levi L Blazer[3], Jarrett J Adams[3], Lingling Zhang[2], Heidi Roehrich[2], Ha-Neul Jo[2], Somasekar Seshagiri[3], Sachdev S Sidhu[3,4,5,*] (iD), Harald J Junge[2,**] (iD) & Stephane Angers[1,3,6,***] (iD)

## Abstract

The FZD4:LRP5:TSPAN12 receptor complex is activated by the secreted protein Norrin in retinal endothelial cells and leads to βcatenin-dependent formation of the blood–retina–barrier during development and its homeostasis in adults. Mutations disrupting Norrin signaling have been identified in several congenital diseases leading to hypovascularization of the retina and blindness. Here, we developed F4L5.13, a tetravalent antibody designed to induce FZD4 and LRP5 proximity in such a way as to trigger βcatenin signaling. Treatment of cultured endothelial cells with F4L5.13 rescued permeability induced by VEGF in part by promoting surface expression of junction proteins. Treatment of $Tspan12^{-/-}$ mice with F4L5.13 restored retinal angiogenesis and barrier function. F4L5.13 treatment also significantly normalized neovascularization in an oxygen-induced retinopathy model revealing a novel therapeutic strategy for diseases characterized by abnormal angiogenesis and/or barrier dysfunction.

**Keywords** blood–retina barrier; endothelial cells; Norrin; retinopathy; Wnt signaling

**Subject Categories** Signal Transduction; Vascular Biology & Angiogenesis

## Introduction

Defects in blood–brain barrier (BBB) and blood–retina barrier (BRB) integrity underlie the ontogeny or progression of a number of diseases such as stroke, pathogen infections, diabetic retinopathy, and neurodegenerative diseases (Obermeier *et al*, 2013; Sweeney *et al*, 2019). Wnt/βcatenin signaling is required for CNS angiogenesis and endothelial cell barrier function during development (Liebner *et al*, 2008; Ye *et al*, 2009; Wang *et al*, 2012) and for the maintenance of the BRB and BBB during postnatal tissue homeostasis (Chang *et al*, 2017; Wang *et al*, 2018). Among the ten vertebrate Frizzled (FZD) receptors, FZD4 is expressed in endothelial cells where it acts as a cell surface receptor for the secreted proteins Norrin (NDP) (Ye *et al*, 2009) and WNT7A/B (Cho *et al*, 2017). Activation of FZD4 leads to βcatenin-mediated regulation of context-dependent genes important for barrier functions, including *Sox17*, *Claudin-5*, and *MFSD2* (Schäfer *et al*, 2009; Ye *et al*, 2009; Wang *et al*, 2020) and thus limits paracellular permeability and endothelial cell transcytosis (Ben-Zvi *et al*, 2014). Norrin plays an especially important role for the development and maintenance of the retinal vasculature (Luhmann *et al*, 2005; Wang *et al*, 2012, 2018). In this context, LRP5 and TSPAN12 function as co-receptors with FZD4 and are required for Norrin signaling (Junge *et al*, 2009). Whereas LRP5 and LRP6 are also obligatory co-receptors for Wnt proteins during βcatenin signaling, TSPAN12 acts as a selectivity gate and signal amplifier exclusively in Norrin-induced signaling (Lai *et al*, 2017). Highlighting the critical roles of Norrin-FZD4 signaling in the retina, *NDP*, *FZD4*, *LRP5,* and *TSPAN12* mutations were identified in a number of related congenital diseases with ocular manifestations, such as Norrie disease, osteoporosis-pseudoglioma syndrome, and familial exudative vitreoretinopathy (FEVR; Warden *et al*, 2007; Baron & Kneissel, 2013; Gilmour, 2015), each of which are associated with hypovascularization of the retina and severe loss of visual function. Overall, disruption of *Ndp* (Richter *et al*, 1998), *Fzd4* (Xu *et al*, 2004), *Lrp5* (Xia *et al*, 2010; Chen *et al*, 2011), or *Tspan12*

1   Leslie Dan Faculty of Pharmacy, University of Toronto, Toronto, ON, Canada
2   Department of Ophthalmology and Visual Neurosciences, University of Minnesota, Minneapolis, MN, USA
3   AntlerA Therapeutics, Foster City, CA, USA
4   Donnelly Centre, University of Toronto, Toronto, ON, Canada
5   Department of Molecular Genetics, University of Toronto, Toronto, ON, Canada
6   Department of Biochemistry, University of Toronto, Toronto, ON, Canada
    *Corresponding author. Tel: +1 416 946 0863; E-mail: sachdev.sidhu@utoronto.ca
    **Corresponding author. Tel: +1 612 624 6017; E-mail: junge@umn.edu
    ***Corresponding author. Tel: +1 416 978 4939; E-mail: stephane.angers@utoronto.ca
    †These authors contributed equally to this work

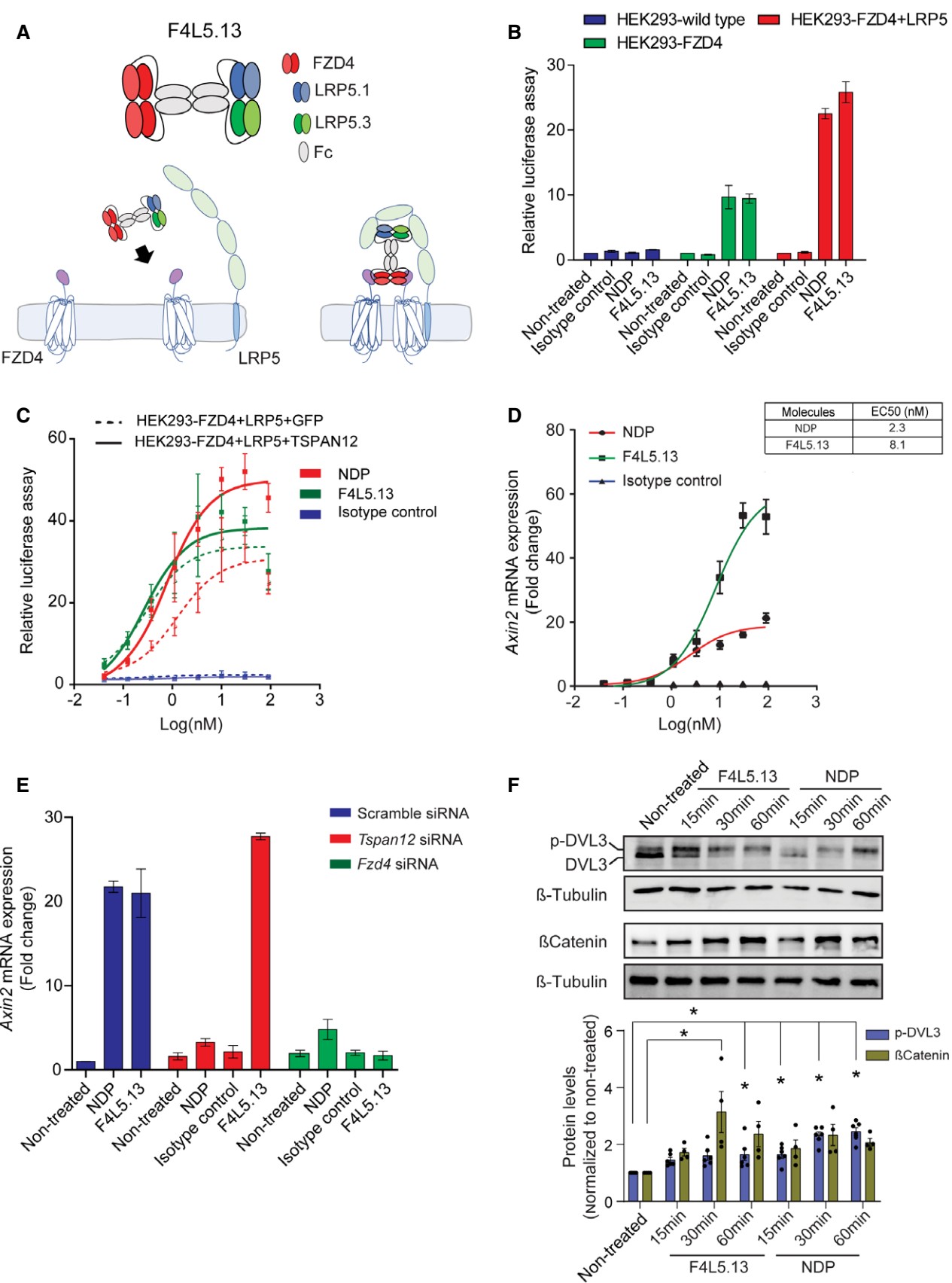

**Figure 1.**

**Figure 1.  F4L5.13 treatment activates the Wnt-βcatenin pathway in LEF/TCF reporter cell lines and cultured endothelial cells.**

A   Top: Molecular architecture of tetravalent F4L5.13. Bottom: Schematic for activation of Wnt/βcatenin signaling by F4L5.13.

B   Activation of βcatenin signaling by F4L5.13 or recombinant NDP (30 nM each) in HEK293T cells transfected with FZD4 and/or LRP5. Values represent fold activation of LEF/TCF reporter gene. Data are presented as mean ± SEM, $n = 3$.

C   Dose–response curves for the activation of a LEF/TCF reporter gene in HEK293T cells transfected with plasmids encoding FZD4, LRP5 and with either GFP or TSPAN12 by serial dilutions of F4L5.13 or NDP proteins (x-axis). Data are presented as mean ± SEM, $n = 3$.

D   RT–qPCR of *Axin2* expression in bEnd.3 cells treated with serial dilutions of F4L5.13, NDP, or isotype control for 24 h. Data are presented as mean ± SEM, $n = 3$.

E   RT–qPCR of *Axin2* in bEnd.3 cells treated with NDP (200 ng/ml), isotype control or F4L5.13 (1,200 ng/ml) and transfected with control, *Tspan12* or *Fzd4* siRNAs. Data are presented as mean ± SD, $n = 3$ technical replicates. Data are representative of two independent experiments.

F   Time course of phosphorylated Dishevelled-3 (p-DVL3) and βcatenin protein levels in bEnd.3 cells treated with 30 nM of F4L5.13 or NDP. Histogram represents the ratio of the DVL3 phosphorylation levels over total DVL3 protein and βcatenin levels over β-Tubulin measured by densitometry of independent experiments. Data are presented as mean ± SEM, $n = 4$–6 (*$P < 0.05$ as compared with NT). Significance was calculated by one-way ANOVA with Bonferroni's multiple comparisons test (*$P < 0.05$ as compared with NT).

(Junge *et al*, 2009; Zhang *et al*, 2018) genes in mice causes similar retinal vascular phenotypes characterized by abnormal blood vessel development and BRB defects.

Importantly, endothelial cell-specific expression of a dominant-negative T-cell factor-4 (TCF4) or a stabilized version of βcatenin is sufficient to, respectively, phenocopy or reverse the *Ndp* and *Fzd4* loss-of-function phenotypes, suggesting that Norrin signaling chiefly functions through regulation of a βcatenin endothelial cell transcriptional program (Zhou *et al*, 2014). Thus, pharmacological activation of FZD4/βcatenin signaling in endothelial cells may represent a novel strategy to promote or restore barrier function by mimicking Norrin and/or WNT7A/B activity and could represent a novel therapeutic opportunity for the treatment of retinal vascular diseases or neurological diseases driven by barrier defects.

## Results

### F4L5.13 activates βcatenin signaling in endothelial cells

We previously described an induced proximity tetravalent antibody platform called FLAg (Frizzled and LRP5/6 Agonist), which promotes the clustering of Frizzled and LRP5/6 co-receptors and thereby mimics the activity of Wnt proteins to activate βcatenin-mediated transcription (Tao *et al*, 2019). Given the genetic evidence supporting the role for FZD4 and LRP5 for the development of the retinal vasculature and maintenance of barrier function, we used antibody fragments to assemble F4L5.13, a highly potent and selective FLAg mediating FZD4 and LRP5 activation. F4L5.13 consists of a diabody formed by two identical paratopes recognizing FZD4 fused to the N-terminus of a heterodimeric Fc, and a diabody composed of two distinct paratopes, respectively, recognizing the first two propellers (E1E2; Wnt-1 binding site) and the membrane proximal two propellers (E3E4; Wnt-3 binding site) of LRP5 fused to

the C-terminus of the Fc (Fig 1A). F4L5.13 interacts with FZD4 with pM affinity, displays single digit nM binding to LRP5, and importantly, does not interact with any of the 9 other vertebrate FZD receptors and is selective for LRP5 over LRP6 (Fig EV1A–C). Thus, in contrast to pervasive activators of βcatenin signaling such as GSK3β inhibitors or purified Wnt proteins, F4L5.13 should lead to βcatenin-mediated transcriptional regulation in only those cells that co-express FZD4 and LRP5.

We first compared F4L5.13 and NDP for their ability to activate βcatenin-signaling using the pBAR reporter, which faithfully monitors LEF/TCF-mediated transcription. Treatment of HEK293T cells (which express only trace amounts of FZD4) with F4L5.13 or NDP required the presence of ectopically expressed FZD4 to detect reporter activation, whereas treatment with WNT3A led to activation in both wild-type cells and when FZD4 was expressed (Figs 1B and EV1D). Supporting these results, flow cytometry experiments detected F4L5.13 binding only in HEK293T cells overexpressing either FZD4 or LRP5 (Fig EV1E). Consistent with the NDP-specific functions described for TSPAN12, co-expression of FZD4 with TSPAN12 led to potentiation of the NDP response but did not affect the activation mediated by F4L5.13 (Fig 1C). We further identified the murine immortalized brain microvasculature cell line bEnd.3 as a highly NDP-responsive cell line. In these cells, F4L5.13 activated expression of the βcatenin target gene *Axin2* with similar potency but with higher efficacy than NDP (Fig 1D). Knockdown of *Tspan12* led to blunting of the NDP response as previously described (Junge *et al*, 2009; Lai *et al*, 2017), but did not affect signaling promoted by F4L5.13 (Figs 1E and EV2A). In contrast, bEnd.3 cells treated with *Fzd4* or *Lrp5* siRNA were largely unresponsive to either NDP or F4L5.13 (Figs 1E and EV2B–D). Both NDP and F4L5.13 similarly led to Dishevelled phosphorylation and βcatenin stabilization (Fig 1F). We conclude that FZD4 and LRP5 clustering triggered by F4L5.13 is sufficient to activate βcatenin signaling in endothelial cells, and as such, F4L5.13 defines a novel class of FZD4-specific agonists.

**Figure 3.  F4L5.13 treatment induces expression of vasculature development and cell adhesion programs in cultured endothelial cells mimicking Norrin.**

A   Volcano plot of gene expression changes in bEnd.3 cells following 24 h of treatment with 30 nM F4L5.13. Genes with significant changes in expression level are highlighted in blue and genes co-regulated by treatment with 30 nM NDP for 24 h are highlighted in red. Statistical analysis was performed using the DESeq2 R package and adjusted *P*-value (Wald test) threshold less than 0.1 was used. Labeled genes are associated with enriched GO biological processes indicated below.

B   Enrichment Map of overrepresented GO biological processes and pathways in genes regulated by F4L5.13. Node sizes represent the number of genes belonging to individual terms (g:SCS FDR adjusted *P*-value < 0.05, gProfiler).

C   Venn diagrams showing overlap in differentially expressed genes following treatment with F4L5.13 or NDP.

D   Heat maps of differentially expressed genes mapping to the indicated biological processes and pathways across treatment groups in RNA-seq experiment. Mean of $n = 3$ samples per group.

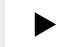

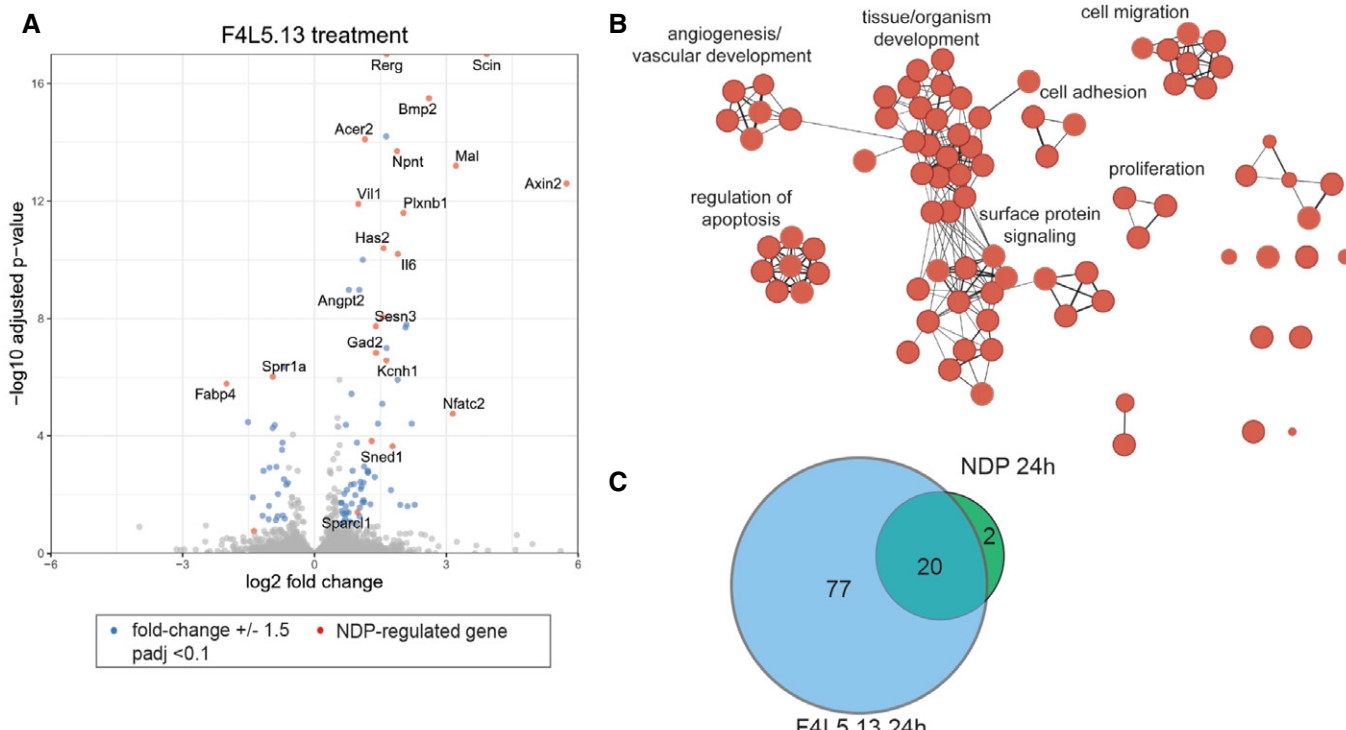

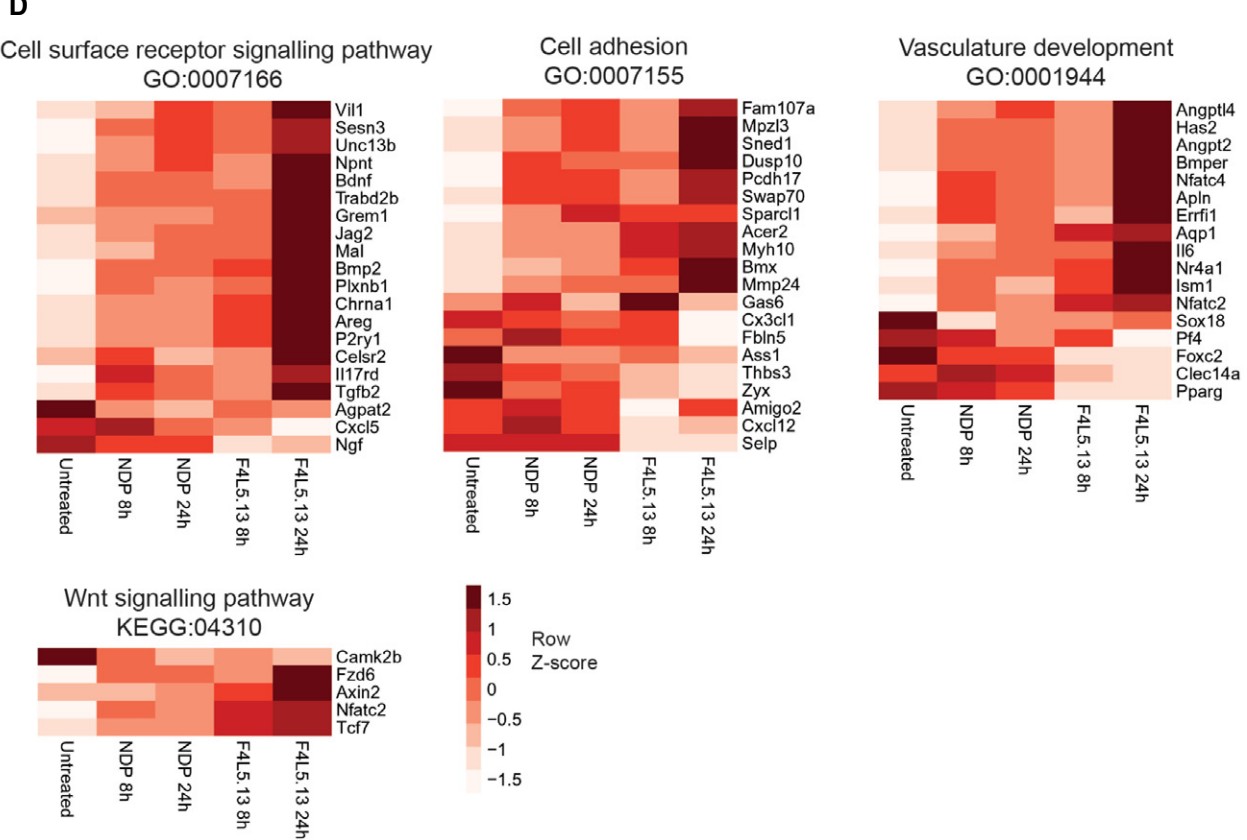

**Figure 2.**

## F4L5.13 mimics NDP treatment in bEnd.3 cells

We next used RNA-seq analysis to identify differentially expressed genes upon treatment of bEnd.3 cells with F4L5.13 or NDP for 8 or 24 h (Fig 2A, Table EV2). Supporting the results above, F4L5.13 treatment strongly induced *Axin2* expression indicating robust activation of βcatenin signaling. Genes differentially expressed following F4L5.13 treatment were enriched for biological processes such as vasculature development/angiogenesis, surface-protein signaling, cell adhesion, and epithelium development (Figs 2B and EV3), consistent with the previously described role of the FZD4-LRP5 signaling axis in CNS vascular development. Importantly, treatment of bEnd.3 cells with NDP for either 8 or 24 h led to differential expression of an overlapping set of genes confirming that F4L5.13 effectively mimics NDP function in endothelial cells (Fig 2A–C). Indeed, following treatment for 24 h, 91% (20/22) of the genes regulated by NDP were also regulated by F4L5.13. A closer examination of expression changes revealed genes linked to GO-Biological Process terms Cell surface receptor signaling pathway (GO:0007166), Cell adhesion (GO:0007155), and Vascular development (GO:0001944) as well as the Wnt signaling pathway (KEGG:04310). When compared with NDP, treatment with F4L5.13 consistently led to increased magnitude of gene induction/repression both by RNA-seq and qPCR validation, which possibly reflects its increased efficacy (Figs 2D and EV4A). Among the genes regulated by both NDP and F4L5.13, *Angpt2* was previously shown to be induced by NDP treatment of cultured human retinal microvascular endothelial cells and to mediate, at least in part, the NDP-mediated increase in proliferation (Ohlmann *et al*, 2010). Another gene strongly induced by both NDP and F4L5.13 was the membrane protein MAL that was previously identified as the receptor for the Clostridium perfringens epsilon toxin (ETX) on endothelial cells and shown to be required for the ETX effects on blood–brain barrier permeability (Linden *et al*, 2019).

## F4L5.13 treatment promotes endothelial cell barrier function

One of the earliest features underlying multiple retinopathies is the Vascular Endothelial Growth Factor (VEGF)-mediated breakdown of the blood–retina barrier and increased endothelial cell permeability. This underlies the current standard of care use of anti-VEGF therapies such as Lucentis (ranibizumab), Eylea (aflibercept), Beovu (brolucizumab), and Avastin (bevacizumab) that revolutionized the clinical management of vascular and exudative diseases of the retina (e.g. macular edema, retinopathy of prematurity, age-related macular degeneration, diabetic retinopathy, retinal vein occlusion, and myopic choroidal neovascularization). However, incomplete responses to anti-VEGF therapy and therapeutic resistance are common clinically and lead to persistent disease and significant unmet clinical needs (Gross *et al*, 2018). Moreover, accumulating evidence indicates that anti-VEGF therapy may not be reversing the nonperfusion underlying pathological diabetic retinopathy (Nicholson *et al*, 2018).

Knowing that FZD4/βcatenin signaling is required for the development and maintenance of CNS endothelial cell barrier properties, we therefore asked whether F4L5.13 treatment could promote barrier functions and oppose the effects of VEGF and determine whether this could represent a novel treatment strategy. Treatment of bEnd.3 cells with F4L5.13 led to increased βcatenin levels and upregulation of ZO-1 expression to comparable levels as NDP (Fig 3A). Then, we investigated the ability of F4L5.13 to rescue endothelial cell barrier breakdown induced by the endothelial cell permeability factor VEGF. When cells were pretreated with VEGF, the endothelial cell barrier was rapidly disrupted, as observed by marked decrease in cell surface localization of tight junction proteins ZO-1, CLDN3, and CLDN5 (Fig 3B). F4L5.13 treatment following VEGF stimulation led to a robust rescue of ZO-1, CLDN3, and CLDN5 cell surface expression indicating that stimulation of FZD4/βcatenin signaling was sufficient to restore tight junction organization disrupted by VEGF treatment (Fig 3B and C). Importantly, in the bEnd.3 cell model βcatenin signaling did not lead to increased *Cldn5* mRNA levels, as was previously shown *in vivo*, or to changes in transcript levels of other cell junction components (Fig EV4B). We conclude that in bEnd.3 cells the effect of F4L5.13 on cell surface levels of junction components is not a result of direct transcriptional regulation. To functionally assess endothelial cell barrier function, we measured the permeability of a confluent endothelial monolayer for 40-kDa FITC-dextran. bEnd.3 cells were grown on transwell filter inserts until confluent and were then treated with VEGF and F4L5.13 simultaneously or were treated with VEGF for 1 h followed by F4L5.13 for 1 h. In both treatment schedules, incubation of bEnd.3 cells with F4L5.13 significantly rescued VEGF-induced permeability (Fig 3D). We conclude from these results that specific activation of the FZD4-LRP5 receptor complex by F4L5.13 promotes cell barrier function, in part, by promoting the assembly of tight junctions.

## F4L5.13 restores retinal vascular development and homeostasis in *Tspan12*$^{-/-}$ mice

We next tested whether F4L5.13 could induce endothelial cell barrier function *in vivo*. To do so, we used *Tspan12*$^{-/-}$ mice with

---

**Figure 3.  F4L5.13 promotes endothelial cell barrier function.**

A   Expression of endothelial junction protein ZO-1 and stabilization of βcatenin in bEnd.3 cells treated with 30 nM F4L5.13 or 30 nM recombinant NDP for the indicated times. Blots are representative of five different experiments. Histogram represents the ratio of ZO-1 and βcatenin levels over β-Tubulin measured by densitometry of five independent experiments. Data are presented as mean ± SEM. Significance was calculated by one-way ANOVA with Bonferroni's multiple comparisons test (*$P < 0.05$ as compared with NT).

B   Immunofluorescence of ZO-1, CLDN3, and CLDN5 localization at bEnd.3 cell junctions. bEnd.3 cells were treated or not with 30 nM F4L5.13 or 30 nM Norrin (NDP) with or without 100 ng/ml VEGF for 1 h. ZO-1 is shown in green, CLDN3/5 in red, and DAPI in blue. Scale bars: 15 μm.

C   Quantification of ZO-1, CLDN3, and CLDN5 fluorescence intensity. Each column represents 40 measurements of fluorescence intensity per condition (y-axis). Data are presented as mean ± SEM. Significance was calculated by one-way ANOVA with Bonferroni's multiple comparisons test (*$P < 0.05$ as compared to VEGF treatment).

D   Transendothelial permeability assay quantifying the passage of FITC-dextran through a monolayer of bEnd.3 cells. Passage of FITC-dextran was measured after exposure of bEnd.3 cells to 100 ng/ml VEGF, 30 nM F4L5.13 or both or pretreated with VEGF for 1 h before treating with F4L5.13 for 1 h. Data are presented as mean ± SD, $n = 5$ independent experiments. Significance was calculated by one-way ANOVA with Bonferroni's multiple comparisons test (*$P < 0.05$ as compared to VEGF treatment).

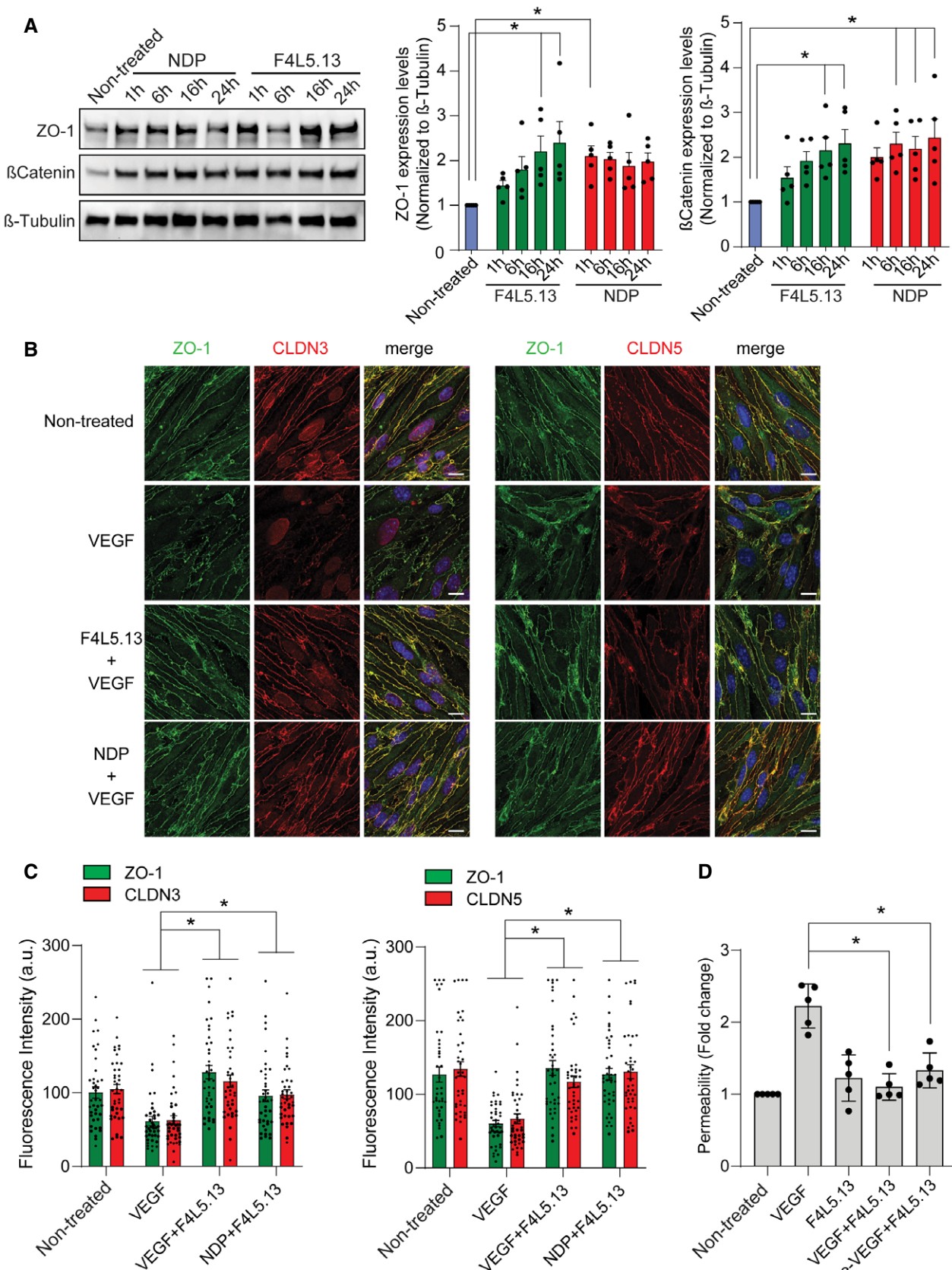

Figure 3.

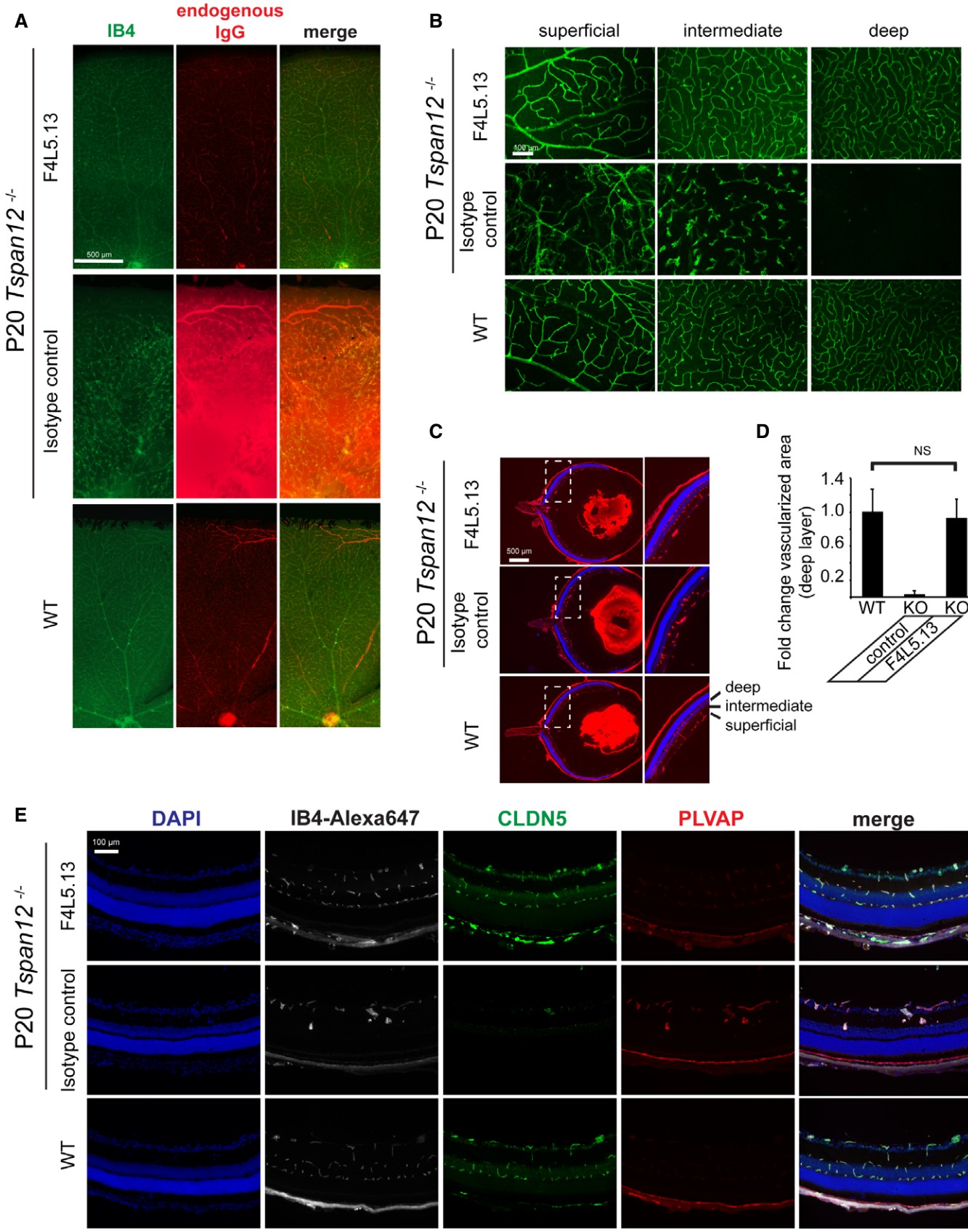

Figure 4.

**Figure 4. F4L5.13 restores intraretinal angiogenesis and blood–retina barrier formation in *Tspan12*$^{-/-}$ mice.**

A   F4L5.13 prevents BRB defects and restores vascular morphogenesis in *Tspan12*$^{-/-}$ mice. 10 mg/kg F4L5.13 or isotype control antibody was injected i.p. every 48 h from P5 to P20. Multiple images acquired at 10× magnification were stitched together to create a high-resolution image of a representative part of the retinal flat mounts. Scale bar 500 μm.
B   Optical sections of IB4-stained P20 retinas showing the mouse three-layered retinal vasculature (superficial, intermediate, and deep layers at 20× magnification) to document blood vessel formation or malformation. In *Tspan12*$^{-/-}$ mice, the intermediate capillary layer fails to properly develop and glomeruloid vascular malformations are present. The deep capillary layer is virtually absent. F4L5.13 fully restores intraretinal capillary development in *Tspan12*$^{-/-}$ mice. Scale bars: 100 μm.
C   IB4-stained retinal cross sections show the restored retinal vasculature and the absence of misdirected angiogenesis after F4L5.13 treatment. Boxed areas are shown enlarged in the panels on the right. Scale bars: 500 μm.
D   Quantification of the vascularized area in the outer plexiform layer. 10 mg/kg F4L5.13 or isotype control antibody was injected every 48 h from P5 to P20. Data are presented as mean± SD, $n$ = 5 retinas (2–4 fields of view per retina) from 4 to 5 mice per group. Significance was calculated by one-way ANOVA with Bonferroni's multiple comparisons test.
E   F4L5.13 rescues barrier function defects in *Tspan12*$^{-/-}$ mice as shown by an increased expression of the tight junction component CLDN5 and decreased expression of the EC fenestration component PLVAP. IB4-Alexa 647 was used to stain ECs. Scale bars: 100 μm.

selectively impaired NDP signaling in the retina, which display retinal vascular morphogenesis defects and impaired endothelial cell barrier functions. *TSPAN12* is one of the genes mutated in FEVR (Gilmour, 2015), and in many respects, *Tspan12*$^{-/-}$ animals recapitulate features observed in FEVR patients. We administered F4L5.13 or an isotype control antibody systemically into developing *Tspan12*$^{-/-}$ mice between P5 and P20, i.e., slightly before and during the time when intraretinal vascular development and barrier formation normally occur. P20 retinas were stained with Isolectin B4 (to reveal endothelial cells and microglia) and anti-mouse IgG (to probe for barrier defects). Stitched high-resolution images (Fig 4A) and optical sections (Fig 4B) were used to document the formation or malformation of the three-layered retinal vasculature. Mice injected with isotype control antibody displayed the characteristic phenotypes of *Tspan12*$^{-/-}$ mice: massive leakage of endogenous IgG through an impaired BRB, absence of intraretinal capillaries in the deep layer, and glomeruloid vascular malformations instead of a properly formed intermediate capillary layer (Zhang *et al*, 2018). Strikingly, BRB function and angiogenesis were virtually completely restored in *Tspan12*$^{-/-}$ mice injected with F4L5.13 (Fig 4A and B), i.e., no substantial IgG extravasation was observed, and intraretinal capillary beds were properly formed. Analysis of retinal cross sections showed that intraretinal capillary formation was restored in the entire retina including central and peripheral regions and that no misdirected angiogenesis occurred (Fig 4C). Quantification of vascular density in the outer plexiform layer (in which the deep capillary bed normally resides) revealed that the FZD4:LRP5 agonist restored intraretinal blood vessel formation in *Tspan12*$^{-/-}$ mice to an extent virtually indistinguishable from wild-type mice (Fig 4D). As expected, the strong BRB defects in *Tspan12*$^{-/-}$ mice were associated with reduction of CLDN5 expression and upregulation of the EC

fenestration component PLVAP. Interestingly, *Tspan12*$^{-/-}$ mice treated with F4L5.13 led to a strong increase in CLDN5 and decreased PLVAP expression (Fig 4E). These findings indicate that F4L5.13 restored FZD4:LRP5 signaling to a level above the threshold required for angiogenesis and barrier formation. Signaling levels may have exceeded endogenous NDP/FZD4 signaling levels, but this did not induce misdirected angiogenesis into avascular layers of the retina or cause an increase of vascular density, perhaps reflecting the fact that Wnt signaling target genes include potent mediators of negative feedback loops. The findings that F4L5.13 activates FZD4:LRP5 signaling independently of TSPAN12 in bEnd.3 cells (Fig 1E) and restores the retinal vasculature in *Tspan12*$^{-/-}$ mice (Fig 4A–E) together confirm the ligand-specific functions of TSPAN12 in NDP/FZD4 signaling.

The striking efficacy of F4L5.13 in the developmental angiogenesis and barrier genesis context prompted us to investigate the effect of F4L5.13 in the adult retinal vasculature characterized by BRB defects. F4L5.13 or isotype control antibody was administered systemically to 2- to 3-month-old *Tspan12*$^{-/-}$ mice, which were then subjected to an acute BRB assay based on quantification of 4 kDa FITC-dextran extravasation during a 1 h time window (Fig 5A). Retinal lysates of *Tspan12*$^{-/-}$ mice treated with the isotype control antibody contained high levels of FITC-dextran tracer, whereas BRB defects were significantly alleviated in *Tspan12*$^{-/-}$ mice treated with F4L5.13 (Fig 5B). This partial restoration of the BRB was not associated with an increase of vascular density (Fig 5C). Furthermore, analysis of *Tspan12*$^{-/-}$ mice injected systemically with sulfo-NHS-biotin, a tracer that forms covalent bonds with proteins and is detected with fluorescent streptavidin probes, confirmed that F4L5.13 partially restored BRB function but had no discernable effect on the aberrant vascular morphology in adult mutant mice (Fig 5D). Specifically, no blood vessels in the

**Figure 5. F4L5.13 partially restored BRB function in mature *Tspan12*$^{-/-}$ mice without any effect on the aberrant vascular morphology.**

A   Injection schema for blood–retina barrier assay in adult *Tspan12*$^{-/-}$ mice with established retinal vascular defects.
B   Quantification of extravasated FITC-dextran in retinal lysates from WT and *Tspan12*$^{-/-}$ mice injected with isotype control antibody or F4L5.13. Data are presented as mean ± SEM, $n$ = 6 retinas from 3 mice per group. Significance was calculated by one-way ANOVA with Bonferroni's multiple comparisons test (*$P$ < 0.05 as compared to *Tspan12*$^{-/-}$ control treatment).
C   Quantification of the vascularized area in adult *Tspan12*$^{-/-}$ mice injected with isotype control or F4L5.13. Data are presented as mean± SD, $n$ = 4 retinas (4 fields of view per retina) from 2 mice per group. $P$-value was calculated by Student's t-test.
D   Flat-mounted retinas of adult *Tspan12*$^{-/-}$ mice injected systemically with sulfo-NHS-biotin show that F4L5.13 partially restored BRB function. PECAM was used to stain endothelial cells. Images are representative of 4 retinas per group. Scale bar: 500 μm.
E   Maximum intensity projection of adult *Tspan12*$^{-/-}$ retinas after mice were injected with F4L5.13 or isotype control. Endothelial cells were stained with IB4-Alexa 647. Images are representative of 4 retinas per group. Scale bar: 100 μm.

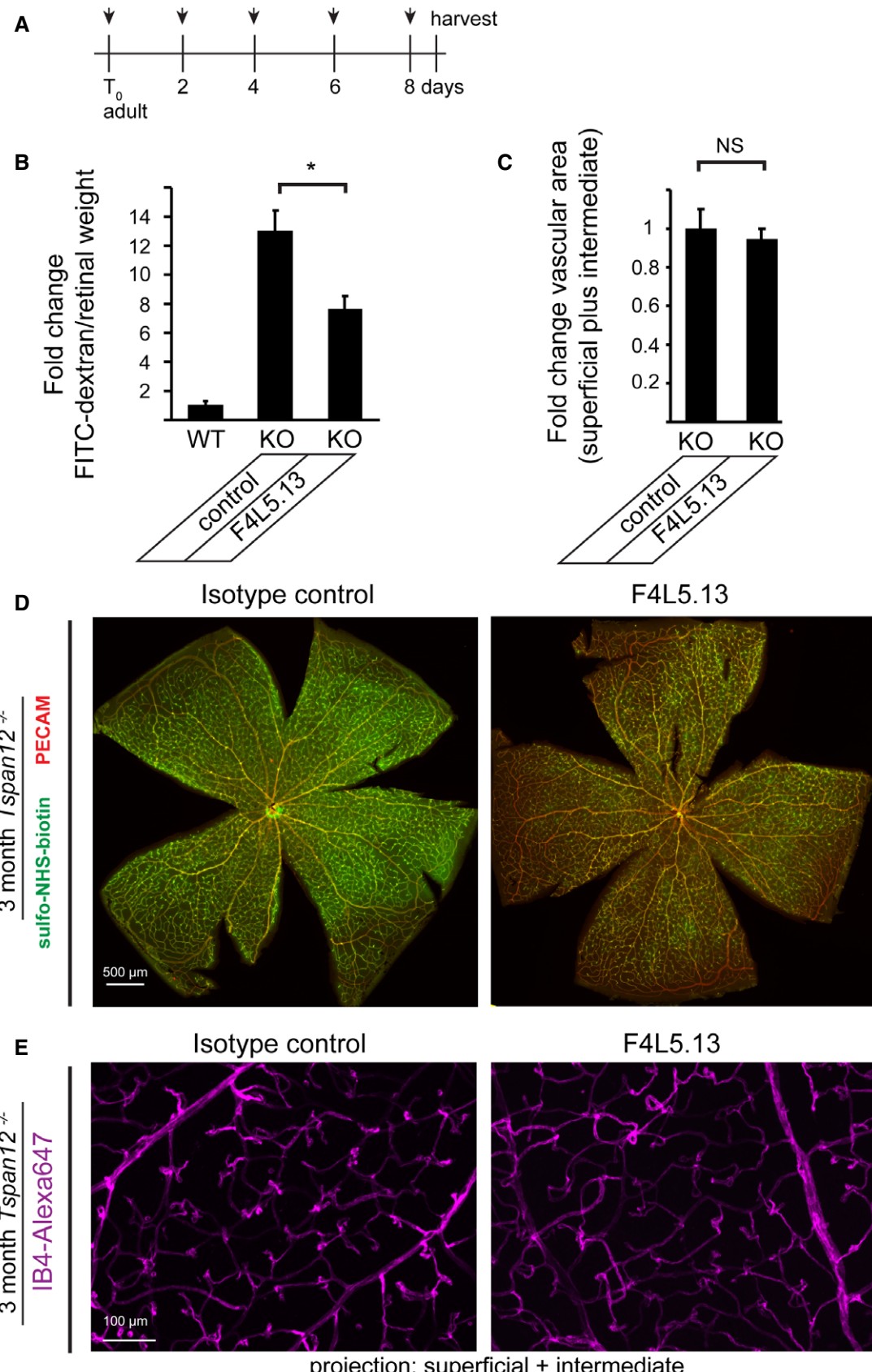

**Figure 5.**

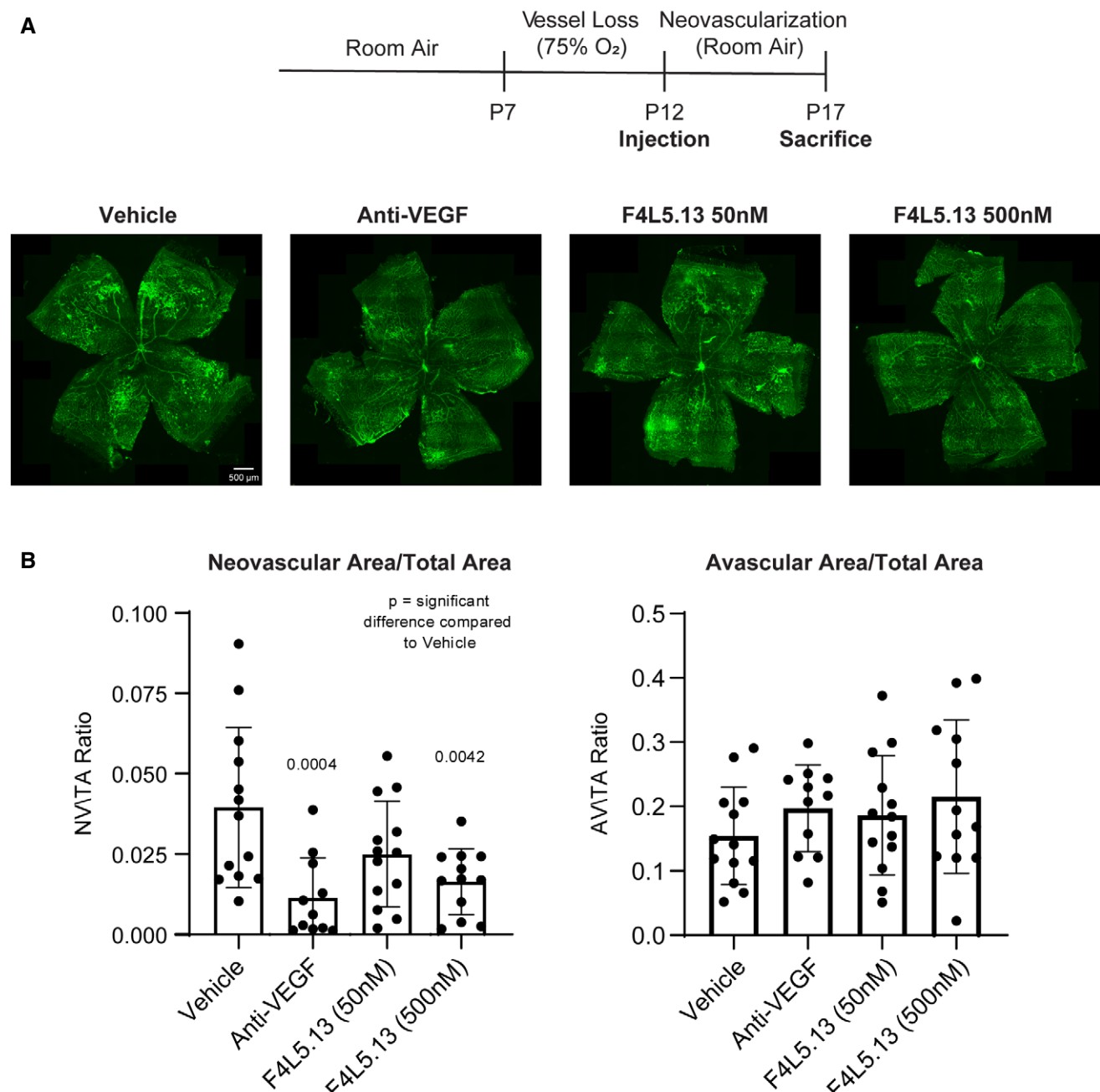

**Figure 6. A FZD4:LRP5 antibody agonist reduces pathological neovascularization in the OIR model.**

A   Schematic diagram of the OIR model (top) and representative images of P17 OIR retinas are shown (bottom). Neonatal mice were exposed to 75% oxygen from P7 to P12 to induce vessel loss. Then mice received intravitreal injections of PBS vehicle, mouse-specific anti-VEGF (as a positive control) delivered at 0.1 µg/µl, or F4L5.13v2 (a second-generation F4L5.13 modality, in which the N-terminal FZD4-specific diabody was replaced with a FZD4-specific Fab) at 50 nM or 500 nM target vitreous concentrations. Mice were returned to room air from P12 to P17 to induce maximum pathologic neovascularization at P17. At P17, retinas from injected mice were collected, dissected as flat-mounts, and stained with Isolectin B4. Scale bar = 500 µm.

B   Quantification of the ratio of neovascular and avascular area on total area of vascularization. Data are presented as mean ± SEM, $n$ = 11–13 retinas per group. Significance was calculated by one-way ANOVA with Bonferroni's multiple comparisons test (indicated $P$-values are in comparison to Vehicle).

outer plexiform layer and no vascular cells with tip cell morphology were observed (Fig 5E). Thus, F4L5.13 partially restored BRB function in mature animals, in which barrier induction had failed during development. Administration of F4L5.13 was not sufficient to induce angiogenesis in the quiescent adult vasculature of $Tspan12^{-/-}$ mice. These results show that blood-CNS barrier function can be modulated by FZD4:LRP5-specific agonists in the mature CNS and that the antibody agonists described herein are a novel class of drug candidates that promote and restore barrier function in pathologically altered CNS vasculatures.

### A FZD4:LRP5 antibody agonist reduces pathological neovascularization in the OIR model

We next evaluated the efficacy of a NDP mimetic FZD4:LRP5 antibody agonist to reduce the formation of pathologic retinal neovascularization (NV) in the oxygen-induced retinopathy (OIR) model, which exhibit pathological features reminiscent of retinopathy of prematurity (ROP) and other human retinopathies such as diabetic retinopathy. For the OIR model, mouse pups were exposed to 75% oxygen from P7 to P12 followed by a return to room air oxygen until P17 to induce retinal ischemia and proliferative vascular disease in the retinal vasculature, at which point they were sacrificed and their retinas isolated. At P12, mice received intravitreal injections of PBS vehicle, mouse-specific anti-VEGF antibodies (positive control), or the FZD4: LRP5 agonist antibodies. Quantification of pathological neovascularization revealed that both treatment groups significantly reduced the neovascular area when compared to vehicle-injected mice (Fig 6A and B). No significant effect on the avascular area was observed in any treatment group. These results in combination with the work detailed above demonstrate clear *in vivo* efficacy of the FZD4:LRP5 agonist in multiple animal models of human retinopathies.

## Discussion

Taken together, our results describe the development of F4L5.13, a selective tetravalent antibody agonist of the FZD4:LRP5 receptor complex. F4L5.13 promotes the proximity of FZD4 with LRP5 co-receptors in such a way as to stabilize a conformation that is likely induced by NDP or Wnt proteins and compatible with downstream signaling activated by these endogenous growth factors. Such molecules extend the emerging landscape of multifunctional drugs functioning through "induced proximity" (Deshaies, 2020) such as molecular glue (e.g. immunomodulatory IMiDs drugs) or proteolysis targeting chimeras (PROTACs) that recruit neosubstrates to ubiquitin ligase scaffolds for degradation.

Our study also describes the bEnd.3 mouse brain endothelial cell line as being Norrin-responsive, capable of mounting a βcatenin transcriptional response in response to NDP that requires the FZD4-LRP5-TSPAN12 receptor complex. bEnd.3 cell line is a Polyoma middle T transformed line and although these cells may provide a powerful *in vitro* system to tease apart signaling mechanisms, an important limitation is the difference in the transcriptional responses observed following treatment with NDP or F4L5.13 and the target genes known to be involved *in vivo*. For example, whereas *Cldn5* is a known βcatenin target gene in endothelial cells *in vivo* (Wang *et al*, 2012; Fig 4E), it is not modulated by NDP or F4L5.13 in bEnd.3 cells (Fig EV4B).

To our knowledge, F4L5.13 is the first biologic that activates Wnt/βcatenin signaling and displays efficacy in multiple mouse models of human disease. In the context of the retinal vasculature, stimulation of FZD4:LRP5 signaling may provide benefits in FEVR or other rare congenital diseases that harbor mutations within Wnt pathway components but could also help mitigate the vascular leakage associated with neovascularization in various clinical settings including diabetic retinopathy and wet macular degeneration that represent clinical unmet needs. Interestingly, F4L5.13 also suppressed neovascularization in the OIR model, possibly reflecting

functional antagonism of VEGF activities. In addition, therapeutic activation of FZD4 could also be beneficial in other vascular diseases of the CNS—such as stroke, traumatic brain injury, multiple sclerosis and Alzheimer's disease—where endothelial cell barrier dysfunction is proposed to contribute to disease initiation and/or progression.

## Materials and Methods

### Cell culture and treatment

bEnd.3 cells, obtained from ATCC (CRL-2299), were grown in DMEM with 4.5 g/l glucose, 3.7 g/l sodium bicarbonate, 4 mM glutamine, 10% FBS, 1% penicillin/streptomycin and cultured in 10 cm dishes until confluency. bEnd.3 cells were passaged using trypsin 0.25%-EDTA solution at a split ratio of 1:4–1:10 and cells were used between p2 and 19 for the experiments. For cell treatment, 200,000 cells were seeded per well of a 6-well plate. After 16 h, cells were around 60–70% confluency and were treated with Norrin (R&D, Cat# 3014-NR), F4L5.13, and isotype control for 24 h. HEK293T cells were authenticated with STR profiling at The Centre for Applied Genomics (Toronto, Ontario, Canada). HEK293T cells were cultured in high-glucose DMEM with 10% FBS at 37°C in the presence of 5% $CO_2$. For maintenance, cells were split 1:6 at near confluence using 0.05% Trypsin-EDTA. Cells were routinely tested for Mycoplasma using the MycoAlert Plus detection kit (Lonza, LT07-701).

### Recombinant Proteins and Reagents

Fc-tagged fusions of FZD1 (5988-FZ-050), FZD2 (1307-FZ-050), FZD4 (5847-FZ-050), FZD5 (1617-FZ-050), FZD7 (6178-FZ-050), FZD8 (6129-FZ-050), FZD9 (9175-FZ-050), FZD10 (3459-FZ-050), Her2 (1129-ER-050), and histidine-tagged fusion proteins of mouse LRP5 (7344-LR-025/CF) and mouse LRP6 (2960-LR-025) were purchased from R&D Systems (Minneapolis, MN, USA). Fc-tagged fusions of the FZD6 ECD were prepared as previously described (Tao *et al*, 2019).

F4L5.13 and the isotype control were cloned and produced as previously described (Tao *et al*, 2019). The Wnt receptor-binding paratopes which comprise F4L5.13 were identified from recombinant selections on mouse LRP5 or human FZD4 using a phage displayed synthetic Fab library (Persson *et al*, 2013) using methods as previously described (Tao *et al*, 2019). For F4L5.13v2, the N-terminal diabody targeting FZD4 was replaced with a Fab specific for FZD4. The isotype control molecule is comprised of paratopes that bind to a non-host protein (gaussia luciferase) that were identified in a similar manner using recombinant phage-display selections.

### Luciferase reporter assay

HEK293T cells were transduced with lentivirus coding for the pBARls reporter (Biechele & Moon, 2008) and with *Renilla* Luciferase as a control to generate a Wnt-βcatenin signaling reporter cell line. To measure F4L5.13 activation, we transduced this cell line with lentivirus coding for human FZD4 protein. For luciferase assay, $2 \times 10^5$ cells were seeded in each well of 24-well plates for 24 h

prior to stimulation. The following day, F4L5.13 protein was added, and following 15–20 h of stimulation, cells were lysed and luminescence was measured in accordance with the dualluciferase protocol (Promega) using an Envision plate reader (PerkinElmer). For TSPAN12 dependency, HEK293T-TopFlash cells were seeded at $0.5 \times 10^6$ cells in a 6-well plate and the next day cells were transfected with FZD4, LRP5 and with either GFP or TSPAN12. Cells were transfected using Lipofectamine 2000 according to the manufacturer's instructions. The next day, cells were passed in a 96-well plate with around 35,000 cells/well, and after 5 h, cells were treated with either F4L5.13 or recombinant NDP overnight. After 16 h, cells were lysed, and luminescence was measured.

### Biolayer interferometry

BLI experiments were performed using an Octet HTX instrument (Forte Bio). FZD selectivity experiments were performed by immobilizing Fc-fused antigen on AHQ BLI Sensors (Cat# 18–5001, ForteBio) to achieve a BLI response of 0.4–0.6 nm. Remaining binding sites on the sensors were then saturated with human Fc (009-000-008, Jackson ImmunoResearch). Antigen or control (Fc alone) coated sensors were transferred to 100 nM of F4L5.13 or isotype control diluted in assay buffer (PBS, 1% BSA, 0.05% Tween-20). Association was monitored for 300 s with a shake speed of 1,000 rpm at a temperature of 25°C. Steady-state response values were determined at 295 s into the association phase. The specific binding signal for each sensor was determined by subtracting the signal from corresponding Fc-coated control sensors.

Affinity measurements of F4L5.13 for FZD4 were performed in a similar manner, but FZD4-coated sensors were transferred to a series of five F4L5.13 dilutions ranging from 25 nM to 1.56 nM. Association was monitored for 300 s after which sensors were transferred to wells containing assay buffer and dissociation was monitored for an additional 300 s with a shake speed of 1,000 rpm at a temperature of 25°C. Kinetic constants were determined by fitting the resulting data were to a 1:1 langmuir model using the Octet Data Analysis software (v11.1.0.4).

Affinity measurements of F4L5.13 for mLRP5 were performed by immobilizing mLRP5 on AR2G sensors (18–5092, ForteBio) to a level of 0.3–0.4 nm via aminocoupling. Remaining amino-reactive sites on the sensors were quenched with ethanolamine and allowed to equilibrate in assay buffer for 30 min. Sensors were then transferred to a series of 7 F4L5.13 dilutions ranging from 100 nM to 0.137 nM. Association was monitored for 300 s after which sensors were transferred to wells containing assay buffer and dissociation was monitored for an additional 300 s. Kinetic constants were determined as described above for FZD4 affinity measurements.

### Flow cytometry

HEK293T cells were seeded at $0.5 \times 10^6$ cells in a 6-well plate and the next day cells were transfected with plasmids encoding for FZD4 or LRP5. Cells were transfected using Lipofectamine 3000 according to the manufacturer's instructions. 48 h after transfection, cells were harvested, washed with PBS and blocked with 1% BSA for 30 min on ice. Then, cells were stained with 30 nM of isotype control or F4L5.13 antibody for 1 h on ice. Cells were washed with PBS and stained with Alexa Fluor 488 Goat anti-human IgG, Fc fragment specific (Jackson ImmunoResearch #109-545-008) and incubated for 30 min at 4°C in the dark. Fixable viability dye eFluor 750 was added and incubated for 10 min, followed by fixation using 4% Paraformaldehyde (PFA). Cells were analyzed by flow cytometry using a Beckman Coulter Cytoflex S instrument (Beckman Coulter, Brea, CA, USA).

### LRP5/6 ELISA assay

LRP5 or LRP6 were coated onto maxisorp 384-well plates at 4°C overnight in PBS at a concentration of 1 μg/ml. Wells were blocked for 1 h at room temperature in PBS supplemented with 0.5% BSA and then washed 6 times in PBST (PBS + 0.05% Tween-20). A series of 8 dilutions of F4L5.13 or a non-targeting antibody (4275) ranging from 100 nM to 0.78 nM in PBST were added to the wells for 1 h at room temperature in a total volume of 30 μl. Unbound protein was removed by washing 6 times with PBST. Thirty microliters of a horseradish peroxidase-conjugated anti-human Fc antibody (Jackson Immuno Cat# 109-036-098) diluted in PBS + 0.5% BSA was added to each well for 30 min at room temperature, after which unbound protein was removed by washing times in PBST. ELISA wells were developed using TMB reagent (Mandel Cat# KP-50-76-03) and quenched with phosphoric acid. Absorbance at 450 nm was determined in an Epoch plate reader (BioTek). Data were analyzed using nonlinear regression in GraphPad Prism (v.9.0.0).

### siRNA transfection

bEnd.3 cells cultured in DMEM with 10% FBS were plated at 100–150 K cells per well of a 12-well culture plate and transfected with 20 nM final concentration of each siRNA, using RNAimax (Invitrogen). The following siRNA were used: Non-targeting control; Qiagen AllStars negative control (order number 1027281), mFzd4 siRNA; 5' – CCUGUUAUUUCUAUGAAAU-dTdT –3' (sense), mTspan12 siRNA; Sigma, SASI_Mm01_00181791, and mLRP5 siRNA; IDT mm. Ri. Lrp5.13.1 48 h later, cells were stimulated with 200 ng/ml recombinant NDP, 1,200 ng/ml of F4L5.13 or isotype control for 24 h. Subsequently, cells were washed three times with PBS, and collected into TRIzol for qPCR analysis. Total RNA was extracted according to the manufacturer's instructions. *Tspan12*, *Fzd4*, and *Lrp5* knockdown efficiency were validated by qPCR (Fig EV2A,B,D).

### qRT–PCR

Total RNA was extracted from bEnd.3 cells using TRIzol reagent (Invitrogen; Thermo Fisher Scientific, Inc., Waltham, MA, USA) according to the manufacturer's protocol and quantified using a nanodrop. Total RNA (1–2 μg) was reverse-transcribed into cDNA using the High-Capacity cDNA Reverse Transcription Kit (Applied Biosystems, Cat#4368813) or Maxima Reverse Transcriptase (Fermentas) according to the manufacturer's protocol. qPCR was performed using the Power SYBR Green PCR Master Mix (Applied Biosystems, Cat# 4368708) using the following thermocycling conditions: 95°C for 10 min, followed by 40 cycles of 95°C for 15 s, 60°C for 1 min, and 72°C for 30 s. Primer sequences indicated in Table EV1 were used, and gene expression was evaluated by the $2^{-\Delta\Delta CT}$ method.

## Immunofluorescence

bEnd.3 cells were cultured on glass coverslips in a 24-well plate. Treated cells were washed with cold PBS and fixed for 20 min with cold 100% methanol at −20°C. Fixed cells were rinsed with PBS and blocked with 1% BSA for 1 h at RT. Cells were incubated for 2 h with primary Abs for ZO-1 ZO-1 (mAb, Invitrogen #339100, 1:100), CLDN3 (pAb, Thermo Fisher 34–1,700; dilution 1:100) or CLDN5 (pAb, Thermo Fisher 34–1,600; dilution 1:100) in 1% BSA. After washing with PBS, cells were incubated for 1 h with Alexa Fluor 488-labeled donkey anti-mouse or Alexa Fluor 568-labeled donkey anti-rabbit antibodies. Coverslips were mounted using Fluoromount (Sigma-Aldrich) with 1% DAPI to stain the nucleus and analyzed on a Zeiss LSM700 confocal microscope using a 60× oil objective. Images were assembled using ImageJ and Photoshop CS6 (Adobe Systems, Mountain View, CA). The fluorescence intensity of ZO-1, CLDN3, and CLDN5 across 40 cell–cell junctions per condition was quantified using ImageJ.

## Endothelial permeability assay

Permeability across endothelial cell monolayers was measured using Nunc polycarbonate transwell units (3.0 μm pore size polycarbonate porous membrane; Thermo Fisher Cat#140627). bEnd.3 cells were seeded at $0.75 \times 10^5$ cells per well and were cultured for 5 days until the formation of a tight monolayer. Before treatment, cells were serum-starved for 3 h in DMEM containing 1% FBS. Then, bEnd.3 cells were treated with either VEGF (100 ng/ml), F4L5.13 (30 nM) or both or pretreated with VEGF for 1 h and then treated with F4L5.13 for 1 h. After treatment, 1 mg/ml FITC-labeled dextran (molecular mass: 40 kDa) was added to the upper chambers for 10 min. Endothelial permeability was measured by collecting 25 μl sample from the lower compartment, which was diluted with 150 μl phosphate-buffered saline (PBS) and measured for fluorescence at 520 nm when excited at 492 nm with a Cytation 5 Cell Imaging Multi-Mode Reader (BioTek, Winooski, VT).

## RNA sequencing

For RNA sequencing, bEnd.3 cells were treated with either F4L5.13 or recombinant NDP for 8 and 24 h. Cells were lysed in TRizol (Thermo Fisher), and RNA was extracted following manufacturer's instructions. Sequencing libraries were prepared and run on Illumina NextSeq-500 instrument at the Lunenfeld-Tanenbaum Research Institute Sequencing Facility (Toronto, ON) generating 75 bp single-read FASTQ files. The transcript reads from FASTQ files were aligned to the Mus musculus transcriptome (Ensembl version GRCm38.96) and quantified using Kallisto (Bray *et al*, 2016), following default parameters. Differential expression analysis was performed using the DESeq2 R package (Love *et al*, 2014) using default parameters with the independent filtering option disabled, which thresholds of raw fold change > 1.5 and adjusted *P*-value < 0.1. Testing for enrichment of biological processes and pathways in differentially expressed genes was performed by querying the Gene Ontology, Reactome, and KEGG datasets using gProfiler (Auffray & De Meulder, 2016). A list of the top 20 most representative GO terms for F4L5.13 regulated genes was generated

using the REVIGO webtool (Supek *et al*, 2011). Heat maps of genes mapping to indicate GO-BP or KEGG terms were generated using the pHeatmap R package.

## Western blot assay

bEnd.3 cells were solubilized with lysis buffer (1% Nonidet P-40, 0.1% sodium dodecyl sulfate (SDS), 0.1% deoxycholic acid, 50 mM Tris (pH 7.4), 0.1 mM EGTA, 0.1 mM EDTA, 20 mM sodium fluoride (NaF), 1:500 protease inhibitors (Sigma), and 1 mM sodium orthovanadate (Na3VO4)). Lysate was incubated for 30 min at 4°C, centrifuged at 14,000 *g* for 10 min, boiled in SDS sample buffer, separated by SDS-polyacrylamide gel electrophoresis, transferred onto a nitrocellulose membrane and Western blotted using indicated Abs: DVL3 (mAb, Santa Cruz #sc8027, 1:1,000), ZO-1 (mAb, Invitrogen #339100, 1:1,000), βcatenin (pAb, CST #9587, 1:5,000), β-Tubulin (mAb, DSHB, 1:5,000). Antibody detection was performed by a chemiluminescence-based detection system (ECL; Thermo Fisher). Quantification of protein levels was done by measuring the band intensities using ImageJ and presented as means of at least four independent experiments.

## Animal

All animal work was done in accordance with the Association for Research in Vision and Ophthalmology statement for the Use of Animals in Ophthalmic and Vision Research and with approval from the Institutional Animal Care and Use Committee at the University of Minnesota or Vanderbilt University Institutional Animal Care and Use Committee. Animals of both genders were used for all studies. The *Tspan12* null allele (*Tspan12*[tm1.2Hjug]) was previously described (Zhang *et al*, 2018). Mice were backcrossed to a C57BL/6J background for more than 10 generations. C57BL/6J mice were purchased from Jackson Labs. WT and *Tspan12*$^{-/-}$ C57BL/6J mice of both sexes were used at the ages indicated in the manuscript text. All mice were housed in a specific pathogen-free facility with a 12 h light–dark cycle at room temperature.

## Analysis of retinal whole mounts using confocal microscopy

F4L5.13 or isotype control antibody (directed against Gaussia luciferase) were administered at 10 mg/kg, i.p., into P5 *Tspan12*$^{-/-}$ pups or adult mice. Injections were repeated every 2–3 days until mice were sacrificed. For P20 mice, eyeballs were removed and fixed for 1 h in 4% PFA at RT. Eyeballs were washed three times with PBS, and retinas were dissected and blocked with 5% goat serum, 0.5% Triton X-100 in PBS for 1 h at RT. Retinas were stained overnight at 4°C with a combination of Griffonia Simplicifolia Isolectin B4 Alexa 488 (1:100, Invitrogen, 121411) and goat anti-mouse IgG-Alexa 555 (1:800, Life Technologies, A21422) in blocking buffer, then washed 6 times for 1 h at RT in PBS, 0.5% Triton X-100, and mounted using Fluoromount-G. For adult mice, 250 μl of 20 mg/ml sulfo-NHS-biotin (Thermo Scientific, Cat.# 21335) in PBS was administered i.p. and allowed to circulate for 60 min. Mice were transcardially perfused with 1% PFA to clear the tracer and start fixation. Dissected eyeballs were immersion fixed in 4% PFA for 90 min at RT. Extravasated dye was detected using Streptavidin-

### The paper explained

#### Problem

Retinal homeostasis requires an intact blood–retina barrier (BRB) and BRB dysfunction is associated with retinal diseases. In addition, retinal homeostasis requires a proper balance of angiogenesis vs. vascular quiescence. In the retina, the secreted ligand Norrin (NDP) binds FZD4 and the co-receptors LRP5 and TSPAN12 and leads to βcatenin-dependent development of the retinal vascular and BRB formation. Mutations in genes important for this pathway lead to rare congenital eye diseases, such as Norrie disease and FEVR, in which the retinal vasculature and BRB is disrupted, causing impaired vision or blindness. More prevalent diseases such as diabetic retinopathy and macular degeneration are also characterized by neovascularization and blood–retina barrier defects. New therapeutic approaches are needed that actively restore BRB function and normalize aberrant retinal vasculatures.

#### Results

We have engineered F4L5.13, a tetravalent antibody, designed to induce FZD4 and LRP5 proximity in order to activate downstream βcatenin signaling. We demonstrated that it is selective for FZD4 and that it efficiently activates βcatenin signaling in mouse brain endothelial cells (bEnd.3 cells) to levels surpassing NDP. F4L5.13 mimics the activity of NDP by activating FZD4 and LRP5 but without requiring TSPAN12. F4L5.13 promotes barrier function in endothelial cells in vitro. In the *Tspan12$^{-/-}$* mouse model that exhibits several features of FEVR, systemic injection of F4L5.13 rescued the blood vessel morphogenesis and BRB defects. Finally, when tested in the oxygen-induced retinopathy model, which exhibits hallmarks of neovascular diseases such as retinopathy of prematurity and diabetic retinopathy, the FZD4:LRP5 antibody agonist was found to normalize the observed pathological neovascularization.

#### Impact

F4L5.13 is a novel synthetic tetravalent antibody that allows the precise (FZD4 and LRP5 specific) and potent activation of βcatenin signaling *in vitro* and *in vivo*. This antibody therefore represents a first-in-class antibody to treat ocular diseases as well as stroke and neurological disorders where endothelial cell barrier function is impaired.

Alexa 488 (Invitrogen), and retinal vessels were stained using Griffonia Simplicifolia IsolectinB4-Alexa 647 (Invitrogen, 132450) or antibody PECAM (1:50, BD Biosciences, 550274). 3D image stacks were acquired using a Olympus Fluoview FV1000 BX2 or Keyence BZX800. For quantifications of vascular density in the outer plexiform layer, image stacks were acquired at 20× magnification. Multiple optical sections—which collectively contained the deep layer capillaries—were combined into a maximum intensity projection, which was subjected to thresholding. The vascularized area (IB4 signal above the threshold) was determined using ImageJ software. Immunostaining of retinal cross sections was performed as described previously (Johnson *et al*, 2015) and stained using anti-CLDN5-Alexa 488 (1:100, Invitrogen, 352588), anti-PLVAP (1:100, BD Bioscience, 550563), and Griffonia Simplicifolia Isolectin B4-Alexa 647 (1:100, Invitrogen, 132450).

### Quantitative blood–retina barrier assay

2- to 3-month-old mice received 5 doses of isotype control or F4L5.13 at 10 mg/kg, i.p., every 48 h. 24 h after the last dose, mice received an i.p. injection of 5 μl/g bodyweight of 50 mg/ml 4 kDa FITC-dextran in PBS (Sigma). After circulating for 80 min, dye was removed from the circulatory system by transcardial perfusion with PBS/Heparin (2 U/ml). Retinas were dissected, weighed, collected into 150 μl PBS, and digested with 100 μg/ml Liberase TL (Roche) at 37 degree for 30 min, with intermittent trituration. Retinal cell suspensions were briefly vortexed and then centrifuged at 20,000 *g* for 20 min at 4°C. The supernatant was collected, and fluorescence was determined using a Synergy HTX (BioTek) plate reader and filters appropriate for fluorescein (excitation 485 ± 10 nm, emission 528 ± 10 nm). The fluorescence signal of wells filled with buffer was subtracted.

### Oxygen-induced retinopathy model

C57/BL6 mouse pups were purchased from Charles River (Wilmington, MA). Pups were born and raised with dams in room air (RA; 20.9%) for 7 days (P0-P7), and then moved to a 75% oxygen atmosphere for 5 days (P8-P12). At P12, mice received intravitreal injections of PBS vehicle, mouse-specific anti-VEGF (delivered at 0.1 μg/μl), or F4L5.13v2 at 50 or 500 nM target vitreous concentrations (0.5 μl injection of 250 and 2.5 μM solutions, respectively). Allocation of mice to treatment/control groups was randomized. Mice were anesthetized with isoflurane and treated with topical proparacaine before injection. Treatment and dosing were randomized between eyes, exposure chambers, and housing cages. Mice were sacrificed on P17 (five days after oxygen removal). Pups were not weighed prior to sacrifice, and however, the weigh, physiology, and behavior of the animals were subjectively observed and no abnormalities in pup health was noticed or attributed to a specific treatment group. Both normal, intraretinal vascular growth and pathological, preretinal neovascularization were assessed in isolectin-B4-stained retinal flat mounts via computer-assisted image analysis. Investigators were blinded during treatment and data collection. Resulting measurements were analyzed by ANOVA and Tukey's multiple comparison test. Data are reported as the ratios of neovascular area to total area (NV/TA) and avascular area to total area (AV/TA).

### Statistics

Data were analyzed using GraphPad Prism software in at least three independent experiments. Graphs represent the mean ± SEM or ± SD. Two-tailed Student's t-test was used to compare two groups. Comparisons between multiple groups were made using one-way ANOVA followed by *post hoc* Bonferroni's multiple comparisons test among groups. $P < 0.05$ was considered statistically significant. Exact *P*-values are listed in Appendix Table S1.

## Data availability

RNA-seq data have been deposited to Gene Expression Omnibus (accession number GSE166764) and are publicly available. All data are available in the main text or the supporting information. URL: https://www.ncbi.nlm.nih.gov/geo/query/acc.cgi?acc = GSE166764.

**Expanded View** for this article is available online.

## Acknowledgements

We are grateful to members of the Sidhu, Junge, and Angers laboratories for helpful discussions and to the Vanderbilt Ophthalmic Contract Research Organization for work on the OIR experiment. This work was supported by grants from the Canadian Institutes of Health Research (MOP-84273 and PJT-175160) to SA, the Ontario Ministry of Research and Innovation (RE07-043) to SSS and SA, the Canadian Institutes of Health Research (MOP-136944) to SSS, the CFREF Medicine by Design (MBDC2-2019-03) to SA and SSS, and the National Institutes of Health (R01 EY024261) to HJJ. HR was supported by NIH P30 EY011374.

## Author contributions

RC, MA, GM, LLB, JJA, and LZ designed and performed experiments. PET, HR, and H-NJ performed experiments. AY analyzed bulk RNA sequencing data. SS provided resources. SA, SSS, and HJJ conceived and supervised the study. SA conceived the study and wrote the manuscript with input from all authors.

## Conflict of interest

LLB, JJA, SS, SSS, and SA are shareholders of AntlerA Therapeutics.

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
