## [Review Process File · EMBO Molecular Medicine]

A Norrin/Wnt surrogate antibody stimulates endothelial cell barrier function and rescues retinopathy

Rony Chidiac, Md Abedin, Graham Macleod, Andy Yang, Pierre Thibault, Levi Blazer, Jarrett Adams, Lingling Zhang, Heidi Roehrich, Ha-Neul Jo, Somasekar Seshagiri, Sachdev Sidhu, Harald Junge, and Stephane Angers

DOI: [10.15252/emmm.202113977](https://doi.org/10.15252/emmm.202113977)

Corresponding authors: Stephane Angers (stephane.angers@utoronto.ca) , Sachdev Sidhu (sachdev.sidhu@utoronto.ca), Harald Junge (junge@umn.edu)

Review Timeline:

Submission Date:	15th Jan 21
Editorial Decision:	5th Feb 21
Revision Received:	2nd Apr 21
Editorial Decision:	28th Apr 21
Revision Received:	14th May 21
Accepted:	17th May 21

Editor: Lise Roth

Transaction Report:

5th Feb 2021

Dear Stephane,

Thank you for the submission of your manuscript to EMBO Molecular Medicine. We have now received feedback from the three reviewers who agreed to evaluate your manuscript. As you will see from the reports below, the referees acknowledge the interest of the study, and are overall supporting publication of your work pending appropriate revisions.

Addressing the reviewers' concerns in full (including additional in vivo model such as oxygen-induced retinopathy or diabetic retinopathy) will be necessary for further considering the manuscript in our journal, and acceptance of the manuscript will entail a second round of review.

EMBO Molecular Medicine encourages a single round of revision only and therefore, acceptance or rejection of the manuscript will depend on the completeness of your responses included in the next, final version of the manuscript. For this reason, and to save you from any frustrations in the end, I would strongly advise against returning an incomplete revision.

When submitting your revised manuscript, please carefully review the instructions that follow below. Failure to include requested items will delay the evaluation of your revision:

2) Individual production quality figure files as .eps, .tif, .jpg (one file per figure).

3) A .docx formatted letter INCLUDING the reviewers' reports and your detailed point-by-point responses to their comments. As part of the EMBO Press transparent editorial process, the point-by-point response is part of the Review Process File (RPF), which will be published alongside your paper.

4) A complete author checklist, which you can download from our author guidelines (<https://www.embopress.org/page/journal/17574684/authorguide#submissionofrevisions>). Please insert information in the checklist that is also reflected in the manuscript. The completed author checklist will also be part of the RPF.

6) Before submitting your revision, primary datasets produced in this study need to be deposited in an appropriate public database (see <https://www.embopress.org/page/journal/17574684/authorguide#dataavailability>). Please remember to provide a reviewer password if the datasets are not yet public. The accession numbers and database should be listed in a formal "Data Availability" section (placed after Materials & Method). Please note that the Data Availability Section is restricted to

new primary data that are part of this study.

"This study includes no data deposited in external repositories"

7) We would also encourage you to include the source data for figure panels that show essential data. Numerical data should be provided as individual .xls or .csv files (including a tab describing the data). For blots or microscopy, uncropped images should be submitted (using a zip archive if multiple images need to be supplied for one panel). Additional information on source data and instruction on how to label the files are available at .

8) Our journal encourages inclusion of *data citations in the reference list* to directly cite datasets that were re-used and obtained from public databases. Data citations in the article text are distinct from normal bibliographical citations and should directly link to the database records from which the data can be accessed. In the main text, data citations are formatted as follows: "Data ref: Smith et al, 2001" or "Data ref: NCBI Sequence Read Archive PRJNA342805, 2017". In the Reference list, data citations must be labeled with "[DATASET]". A data reference must provide the database name, accession number/identifiers and a resolvable link to the landing page from which the data can be accessed at the end of the reference. Further instructions are available at .

9) We replaced Supplementary Information with Expanded View (EV) Figures and Tables that are collapsible/expandable online. A maximum of 5 EV Figures can be typeset. EV Figures should be cited as 'Figure EV1, Figure EV2" etc... in the text and their respective legends should be included in the main text after the legends of regular figures.

- Additional Tables/Datasets should be labeled and referred to as Table EV1, Dataset EV1, etc. Legends have to be provided in a separate tab in case of .xls files. Alternatively, the legend can be supplied as a separate text file (README) and zipped together with the Table/Dataset file. See detailed instructions here: .

10) The paper explained: EMBO Molecular Medicine articles are accompanied by a summary of the articles to emphasize the major findings in the paper and their medical implications for the non-specialist reader. Please provide a draft summary of your article highlighting

11) For more information: There is space at the end of each article to list relevant web links for further consultation by our readers. Could you identify some relevant ones and provide such

information as well? Some examples are patient associations, relevant databases, OMIM/proteins/genes links, author's websites, etc...

12) Every published paper now includes a 'Synopsis' to further enhance discoverability. Synopses are displayed on the journal webpage and are freely accessible to all readers. They include a short stand first (maximum of 300 characters, including space) as well as 2-5 one-sentences bullet points that summarizes the paper. Please write the bullet points to summarize the key NEW findings. They should be designed to be complementary to the abstract - i.e. not repeat the same text. We encourage inclusion of key acronyms and quantitative information (maximum of 30 words / bullet point). Please use the passive voice. Please attach these in a separate file or send them by email, we will incorporate them accordingly.

Please also suggest a striking image or visual abstract to illustrate your article. If you do please provide a png file 550 px-wide x 400-px high.

13) As part of the EMBO Publications transparent editorial process initiative (see our Editorial at <http://embomolmed.embopress.org/content/2/9/329>), EMBO Molecular Medicine will publish online a Review Process File (RPF) to accompany accepted manuscripts.

In the event of acceptance, this file will be published in conjunction with your paper and will include the anonymous referee reports, your point-by-point response and all pertinent correspondence relating to the manuscript. Let us know whether you agree with the publication of the RPF and as here, if you want to remove or not any figures from it prior to publication.

I look forward to receiving your revised manuscript.

With my best wishes,

Lise

Lise Roth, PhD
Editor
EMBO Molecular Medicine

To submit your manuscript, please follow this link:

Link Not Available

Photos 400-800 DPI

*Additional important information regarding figures and illustrations can be found at <https://bit.ly/EMBOPressFigurePreparationGuideline>

***** Reviewer's comments *****

Referee #1 (Comments on Novelty/Model System for Author):

Please see the comment for authors. Briefly we find nothing new for the mechanism.

Referee #1 (Remarks for Author):

In the present study, Chidiac and colleagues developed a synthetic antibody (F4L5.13) inducing FZD4/LRP5 clustering and thereby activating b-catenin in vascular endothelial cells. It is claimed that F4L5.13 mimicked NDP treatment promoting barrier function in cultured endothelial cells and abrogated the phenotypes in retinal vasculature in Tspan12^{-/-} mice. Although this reviewer finds no conceptual advance for the molecular pathway and related in vivo phenotypes, the possible utility of this antibody for ocular diseases characterized by defective BRB functions may be of interest to ophthalmologists. Having said that, several points should be addressed as below:

1. (Fig.1F) Kinetics for DVL3 phosphorylation is not exactly the same between F4L5.13 and NDP. Please elaborate more and explain the cause of this difference. Also quantification should be provided
2. (Fig.3A) In the same line as above, the time course of ZO-1 looks different. However, in contrast to Fig. 1f, the effect of F4L5.13 is slower than NDP. Please discuss and provide quantification.
3. (Fig. 2C, D) There seems to be a substantial difference between NDP and F4L5.13 treatment. Please confirm the expression of several key genes by RTqPCR.
4. (Fig.2D) The authors should list the top 10 or 20 significant GO terms in a nonarbitrary way. It is not sufficient to show Go terms authors are interested in.
5. (Fig. 4A) I don't think this introductory schema is necessary. We can find this everywhere.

6. (Fig. 4B) The authors evaluated the BRB function by visualizing injected antibodies. Why can they compare the leakage of different antibodies, F4L5.13 and isotype IgG? Do they have the same molecular weight? Also, it is strange that injected F4L5.13 is not visible inside the vascular lumen at all.

7. (Fig. 4D) Please show the retinal images for FITC-dextran experiments in adult.

8. (Fig. 4C, D) Please explain post-test after one-way ANOVA.

Referee #2 (Comments on Novelty/Model System for Author):

additional models are suggested in the review

Referee #2 (Remarks for Author):

In this manuscript, the authors report the development of a novel tetravalent antibody F4L5.13 that induces the FZD4-LRP5 receptor complex proximity to activate the beta catenin signaling in endothelial cells. The results show that F4L5.13 improved tight junction formation and reduced VEGF-induced permeability in cultured endothelial cells. Moreover, F4L5.13 treatment successfully restored retinal angiogenesis and barrier function in Tspan12^{-/-} mice. While these initial findings are exciting, in particular the rescuing effect of F4L5.13 on angiogenesis in Tspan12^{-/-}, additional studies are needed to validate the effect of F4L5.13 on beta catenin-dependent tight junction protection and its potential use in disease models.

Major points:

1. Figure 1: 1) There is no evidence to demonstrate the direct binding of F4L5.13 to FZD4 and LRP5 on the cell membrane of HEK293T cells or bEnd.3 cells. Including this data could significantly strengthen the molecular basis of study. 2) Please provide statistical significance for comparisons in all graphs. 3) Please provide quantitative data for p-DVL3, DVL3, and beta catenin. In addition to total beta catenin, the level of active beta catenin should be examined. 4) The title is inaccurate since most data were generated in HEK293T cells.

2. Figure 2: 1) The title needs to be revised to better reflect the RNA-seq study in cultured endothelial cells. 2) Key differentially expressed genes identified from RNA-seq should be validated by qPCR. 3) Genes involved in endothelial barrier function were not identified?

3. Figure 3: 1) Fig. 3A: The levels of tight junction proteins, ZO-1, claudin-3, claudin-5, and total and active beta catenin, should be quantified. Were the gene expression levels affected? 2) Fig. 3B: Please quantify the tight junction protein changes, and discuss whether F4L5.13 protected the barrier function in VEGF-treated cells through enhancing protein expression or barrier formation or both. 3) Fig. 3C: Treatment with VEGF or F4L5.13 for only 1 hour seems to be too short to induce changes in tight junction protein expression. Other mechanisms may be involved. In addition, it would be helpful if the permeability change can be validated by TEER measurements. 4) It is also critical to determine whether the protective effect of F4L5.13 on barrier function and formation is through the beta catenin signaling.

4. Figure 4: 1) Fig. 4C. Please specify how many mice in each group were used in the statistical analysis. 2) In Fig. 4B experiment, does restoring vasculature also improve vascular barrier function? 3) In Fig. 4D, while reducing vascular permeability, did F4L5.13 treatment exert any effect on vasculature? 4) In both experiments, please determine the changes in endothelial tight junctions in

retinal vasculature.

5. It would greatly improve the significance of the study by including additional animal models to investigate the effect of F4L5.13 treatment on retinal vascularization and permeability in oxygen-induced retinopathy and diabetic retinopathy.

Referee #3 (Comments on Novelty/Model System for Author):

The model systems applied are to some extent inadequate; please see comments to authors. I suggest to replace data in Fig. 3 with a disease model such as oxygen-induced retinopathy. The number of observations seems small (number of fields of vision is given, but from how many independent retinas?). siRNA efficacy is not shown. Blots are not quantified.

Referee #3 (Remarks for Author):

In this study, the authors have developed a tetravalent antibody agonist (F4L5.13) of the Norrin/Wnt signalling pathway. F4L5.13 was shown to ablate VEGFA-induced endothelial cell permeability in bEND.3 cells. In Tspan^{-/-} mice, used to model impaired retinal vascular development and barrier function, F4L5.13 administration (i.p.) restored post-natal angiogenesis and barrier function in the retina. The effects of the F4L5.13 antibody are impressive, however, some of the in vitro data are questioned (see below) and in other analyses, insufficient information is provided. Specific comments/questions:

1. Blots shown throughout need to be quantified and shown as normalized values relative to the loading control.
2. Please show the efficacy of all siRNA knockdowns.
3. The number of observations should be clarified. Fields of vision should be shown for number of eyes (one eye/mouse). Overall, the n seems small; for example, in Fig. 4D, n=4. Please clarify that this is 4 mice and not 4 eyes from 2 mice.
4. In Fig 1F, the levels of pDVL are considerable in the untreated control and the effect of F4L5.13 at 60 min is similar to the base line, while in the Norrin-treated cells, pDVL levels at 15 min decrease and then perhaps increase again at 60 min (quantification is needed). To make a firm conclusion about kinetics as is done on page 7, first para: "Both Norrin and F4L5.13 led to Dishevelled phosphorylation and bcatenin stabilization with similar kinetics (Fig. 1F).", is not possible. The endothelial cell line used here and in other experiments, b.End.3, is polyoma mT transformed and has a high basal activity. It would be preferable to use primary endothelial cells.
5. In RNA-seq experiments, what was the effect of F4L5.13 treatment on expression of tight junction proteins (i.e. Claudin-5, ZO-1 and others)? This should be included. Please compare with the effect of Norrin, as normalized expression data.
6. The overlap of genes induced by F4L5.13 and Norrin was overlapping only to a small extent (77 vs 20) although the authors seem to conclude the majority was overlapping. Please provide a supplemental list of induced genes for the two treatments at the 8 and 24 h time points. Moreover, F4L5.13 treatment appears to show comparable effects with that of Norrin at 8h but at the 24h time points, considerably higher levels of expression of several genes is evident. Was this a dose dependent effect of F4L5.13?
7. The authors make the claim that VEGF breaks down the blood-retinal barrier by disrupting tight junction organization (page 8 and 9). However, VEGF-induced leakage of macromolecules is known to depend on breakdown of adherens junctions (VE-cadherin/b-catenin), not tight junctions. It is possible that VEGF treatment of in vitro cultures in which tight junctions may not have become

established (due to the transformed nature of the cells or other aspects) may lead to a turnover of Cldn5 without directly reflecting the quality of the barrier. Importantly, in retinopathies, it is very unlikely that VEGF expression represents the trigger of the barrier breakdown (as is correctly described by the authors on page 9, first para). This does not exclude that anti-VEGF therapy is beneficial by suppressing macromolecular leakage and by improving the perfusion by normalizing the vasculature. Thus, while the data shown in Fig 3 are consistent with literature on that VEGF treatment of endothelial cell lines under certain circumstances can reduce Cldn5 expression, this has not been validated in vivo and the relevance for conclusions on the effect of F4L5.13 on e.g. retinopathy is questionable. It would be best to take the data in Fig. 3B and C out and replace with an in vivo model, such as oxygen-induced retinopathy. At the absolute minimum, the authors need to explain how F4L5.13/NDP co-treatment with VEGFA for only 1 hour (too short to allow transcription/translation) preserves CLDN-5 expression the endothelium.

8. According to the literature, Tspan12^{-/-} retinas have a moderate delay in outgrowth of the superficial layer, however, the vertical sprouts and the OPL layer does not form at all. It is not possible from the images in Fig. 4B to estimate the % vascularized area in the Tspan12^{-/-} retinas. This is important as the leakage should be normalized to the vascular area rather than to the weight of the retinas as is done here. Please show the extent of vascularization in the three layers in high and low magnified images.

9. The Tspan12^{-/-} is an interesting model to demonstrate the effect of F4L5.13, however, it is a genetic model rather than a disease model. It would be both important and very valuable if the authors complement their study with an in vivo retinopathy model.

Minor

10. The format of author names in the reference section is inconsistent. In some instances they include first author's name followed by "et al."

***** Reviewer's comments *****

Referee #1 (Remarks for Author):

In the present study, Chidiac and colleagues developed a synthetic antibody (F4L5.13) inducing FZD4/LRP5 clustering and thereby activating b-catenin in vascular endothelial cells. It is claimed that F4L5.13 mimicked NDP treatment promoting barrier function in cultured endothelial cells and abrogated the phenotypes in retinal vasculature in Tspan12^{-/-} mice. Although this reviewer finds no conceptual advance for the molecular pathway and related in vivo phenotypes, the possible utility of this antibody for ocular diseases characterized by defective BRB functions may be of interest to ophthalmologists. Having said that, several points should be addressed as below:

1. (Fig.1F) Kinetics for DVL3 phosphorylation is not exactly the same between F4L5.13 and NDP. Please elaborate more and explain the cause of this difference. Also quantification should be provided

The reviewer is correct that this may have been an overstatement on our part. Although we have shown the time-dependent increase in DVL3 phosphorylation, we had not carefully examined its kinetic. We are now providing quantification of the data in the original manuscript as well as additional experiments. All reviewers had similar requests. Although the activation of DVL3 phosphorylation by NDP and F4L5.13 is overly similar, differences exist in the overall magnitude of phosphorylation as well as in the timing. We have modified the text and removed the emphasis on kinetic.

2. (Fig.3A) In the same line as above, the time course of ZO-1 looks different. However, in contrast to Fig. 1f, the effect of F4L5.13 is slower than NDP. Please discuss and provide quantification.

We provide quantification for increased in ZO-1 seen with NDP and F4L5.13.

3. (Fig. 2C, D) There seems to be a substantial difference between NDP and F4L5.13 treatment. Please confirm the expression of several key genes by RTqPCR.

The reviewer is correct in noting the differences. The main difference is that the magnitude of gene expression changes is greater with F4L5.13 than with NDP. As a result, more differentially expressed genes reach statistical significance with F4L5.13 and most of the gene expression changes seen with NDP are also seen with F4L5.13 (Figure 2C). As requested by the reviewer, we now provide validation by qPCR for several of the genes (Figure EV4A).

4. (Fig.2D) The authors should list the top 10 or 20 significant GO terms in a nonarbitrary way. It is not sufficient to show Go terms authors are interested in.

As suggested by the reviewer we now present the top GO terms in Figure EV3.

5. (Fig. 4A) I don't think this introductory schema is necessary. We can find this everywhere.

We have removed the schematic and show each vascular layer in individual panels instead of color depth projections.

6. (Fig. 4B) The authors evaluated the BRB function by visualizing injected antibodies. Why can they compare the leakage of different antibodies, F4L5.13 and isotype IgG? Do they have the same molecular weight? Also, it is strange that injected F4L5.13 is not visible inside the vascular lumen at all.

We apologize if this was not clear. What is imaged in this figure to assess permeability is the circulating mouse IgGs. Indeed, in wild-type animals the blood retina barrier is intact, and no detectable leakage of IgG is observed. However, in the *Tspan12* KO, the BRB is defective and there is substantial leakage of mouse IgG in the retina. Treatment of the mice with the FZD4 agonist F4L5.13 leads to restoration of the BRB and prevents leakage of the circulating mouse IgG. The figure was modified to highlight that endogenous IgG was stained and make this point more obvious.

7. (Fig. 4D) Please show the retinal images for FITC-dextran experiments in adult.

The FITC-dextran experiment is a quantitative blood-retina barrier assay in which retinas are lysed and the dye is extracted for quantification. Therefore, in the original manuscript no imaging data was presented. However, to address the request of the reviewer, we repeated the experiment and performed sulfo-NHS-biotin injections and anti-PECAM vascular staining to analyze leakage, vascular morphology, and vascular density. Sulfo-NHS-biotin leakage was reduced in all F4L5.13-treated specimens (Fig 5D), confirming the quantitative data that indicated partial restoration of barrier function in adult *Tspan12* KO mice. Analysis of vascular morphology in treated and non-treated adult *Tspan12* KO mice revealed no detectable change in vascular morphology (Fig 5E). In each group the mutant vasculature was aberrantly localized in the space between nerve fiber layer and inner plexiform layer. No active angiogenesis, presence of tip cells, or misdirected angiogenesis was observed. The vascular density was not significantly different in the two groups.

8. (Fig. 4C, D) Please explain post-test after one-way ANOVA.

Comparisons between multiple groups were done using one-way ANOVA followed by *post-hoc* Bonferroni's multiple comparisons test among groups.

Referee #2 (Remarks for Author):

In this manuscript, the authors report the development of a novel tetravalent antibody F4L5.13 that induces the FZD4-LRP5 receptor complex proximity to activate the beta catenin signaling in endothelial cells. The results show that F4L5.13 improved tight junction formation and reduced VEGF-induced permeability in cultured endothelial cells. Moreover, F4L5.13 treatment successfully restored retinal angiogenesis and barrier function in *Tspan12*^{-/-} mice. While these initial findings are exciting, in particular the rescuing effect of F4L5.13 on angiogenesis in *Tspan12*^{-/-}, additional studies are needed to validate the effect of F4L5.13 on beta catenin-dependent tight junction protection and its potential use in disease models.

We thank the reviewer for this assessment and for the insightful suggestions that contributed to make our manuscript stronger.

Major points:

Figure 1:

1) There is no evidence to demonstrate the direct binding of F4L5.13 to FZD4 and LRP5 on the cell membrane of HEK293T cells or bEnd.3 cells. Including this data could significantly strengthen the molecular basis of study.

We thank the reviewer for raising this point. The FZD4 and LRP5 antibodies that we developed to build the F4L5.13 modality have been extensively characterized and are very specific for their target (Fig. EV1A and C). Highlighting the specific requirement for FZD4 and LRP5: 1) F4L5.13 is unable to stimulate β catenin signaling in HEK293 cells, which do not express FZD4 and only low levels of LRP5 (Fig.1B). However, upon FZD4 and LRP5 overexpression, F4L5.13 treatment leads to stimulation of β catenin signaling in HEK293 cells (Fig. 1B) 2) RNAi depletion of *Fzd4* in bEND3 endothelial cells led to robust inhibition of *Axin2* expression (β catenin target gene) stimulated by F4L5.13 (Fig. 1E). As additional data, we now show that knockdown of *Lrp5* also leads to blunting of the F4L5.13-dependent stimulation (Fig. EV2C).

To directly address the suggestion of the reviewer, we now present flow cytometry data (Figure EV1 E) showing binding of F4L5.13 at the surface of HEK293 cells overexpressing FZD4 or LRP5, whereas the modality shows negligible binding to control cells.

2) Please provide statistical significance for comparisons in all graphs. 3) Please provide quantitative data for p-DVL3, DVL3, and beta catenin. In addition to total beta catenin, the level of active beta catenin should be examined.

We have added quantification and statistical analysis for all the graphs.

4) The title is inaccurate since most data were generated in HEK293T cells.

We have modified the title of the figure.

Figure 2:

1) The title needs to be revised to better reflect the RNA-seq study in cultured endothelial cells.

We have modified the title of the figure.

2) Key differentially expressed genes identified from RNA-seq should be validated by qPCR.

As requested by the reviewer we have now validated a panel of differentially expressed genes identified by RNAseq using qPCR (Figure EV4 A).

3) Genes involved in endothelial barrier function were not identified?

The reviewer is correct. Our RNAseq analysis has not identified significant differentially regulated genes involved in barrier function upon treatment of Bend3 cells with NDP nor F4L5.13. This is in contrast to the situation *in vivo*, where FZD4-bcatenin signaling in endothelial cells is known to regulate *Cldn5* expression, a structural component of tight junctions (PMID: 23217714 and this new data added in this manuscript (Figure 4E)). It is thus important to note that although bEnd3 cells strongly respond to Norrin in a FZD4, TSPAN12 and LRP5 dependent manner (and as a result represent a useful tool to study signaling mechanisms mediated by endogenous receptors), differences exist with the *in vivo* context in terms of gene expression and possible mechanisms affecting barrier function. We have modified the text to illustrate these differences. We would like to thank the reviewer for pointing this out, which led us to add this limitation.

Figure 3:

1) Fig. 3A: The levels of tight junction proteins, ZO-1, claudin-3, claudin-5, and total and active beta catenin, should be quantified. Were the gene expression levels affected?

As requested by the reviewer, we are now providing quantification in the cell surface level of the junction proteins. As mentioned above, the expression of these genes was not found to be regulated in the RNAseq experiment. We have directly confirmed this using qPCR (Figure EV4B).

To further address this question in the relevant in vivo context, we directly investigate their expression in the mouse retina of wild-type and *Tspan12*^{-/-} animals before and after treatment with F4L5.13. Claudin3 is not expressed in the mouse retina. Consistent with previously published results, treatment of *Tspan12* KO mice with F4L5.13 led to a marked increase of CLDN5 expression and decreased PLVAP expression, indicating rescue of barrier function.

2) Fig. 3B: Please quantify the tight junction protein changes, and discuss whether F4L5.13 protected the barrier function in VEGF-treated cells through enhancing protein expression or barrier formation or both.

We have now quantified the change in cell surface expression of tight junction proteins (Figure 3C). We surmise from our results that barrier function is promoted due to change in cell surface expression of the junction proteins.

3). Fig. 3C: Treatment with VEGF or F4L5.13 for only 1 hour seems to be too short to induce changes in tight junction protein expression. Other mechanisms may be involved. In addition, it would be helpful if the permeability change can be validated by TEER measurements.

We of course can't exclude other mechanisms than transcriptional regulation. As stated above, the expression of the junctional protein themselves don't seem to be regulated. Our future work is now focused to delineate the molecular mechanisms directing the change in permeability. A possible non-transcriptional mechanism may be effects of stabilized beta-catenin on adherens junctions, which indirectly stabilize tight junctions.

4) It is also critical to determine whether the protective effect of F4L5.13 on barrier function and formation is through the beta catenin signaling.

We attempted to tackle this by using RNAi for β catenin, however we anticipated that this would be difficult given the known role of β catenin at the junction. The experiment confirmed this limitation, cells with β catenin knockdown showed increased permeability, preventing us from studying its requirement. The requirement of β catenin for barrier function downstream of FZD4 activity has been previously established. As noted above, there are important limitations in the bEnd3 system since the junction components are not transcriptionally regulated,

Figure 4: 1) Fig. 4C. Please specify how many mice in each group were used in the statistical analysis.

We included additional mice. The quantification of vascular density at P20 (new Figure 4D) is now quantified from N=5 retinas from 4-5 mice (4 mice in the isotype control group, 5 mice in the other groups.) Figure legends were updated.

2) In Fig. 4B experiment, does restoring vasculature also improve vascular barrier function?

The effects of Norrin signaling on angiogenesis and barrier function can be separated. Loss of Norrin signaling after late-induced endothelial cell-specific deletion of *Tspan12* in 4-week old mice causes strong barrier defects but no angiogenesis defects, as *Tspan12* deletion is induced when retinal vascular development is already completed (Zhang et al., 2018 PMID 30354230). Thus, Norrin signaling is required to maintain the barrier even in normally developed blood vessels. In our experiment in Figure 4 we cannot determine to what extent the rescue of barrier function depends on the rescue of vascular morphogenesis. However, to address the reviewer point we now show that F4L5.13 induces the expression of CLDN5 and partially suppresses PLVAP in mutant blood vessels (new Figure 4E). In addition, we observe partial restoration of barrier function in adult F4L5.13-treated *Tspan12* KO mice, without a corresponding normalization of the aberrant mutant vasculature (new Figure 5). Together, the data suggest that restoring vascular architecture and density is not an absolute prerequisite for restoring barrier function (at least partially).

3) In Fig. 4D, while reducing vascular permeability, did F4L5.13 treatment exert any effect on vasculature?

Please see detailed response to reviewer 1, point 7. In brief, treatment of adult *Tspan12* KO mice with F4L5.13 partially restores barrier function but does not normalize the aberrant mutant vasculature or induce angiogenesis (new Figure 5). It appears that Norrin signaling alone is not sufficient to activate the quiescent adult vasculature, at least not after 8 days of multiple high dose treatments.

4) In both experiments, please determine the changes in endothelial tight junctions in retinal vasculature.

We performed the experiment requested by the reviewer during development at p20. We show that treatment with F4L5.13 leads to a marked increase in CLDN5 expression and partial suppression of PLVAP expression, indicating rescue of barrier functions. We did not have a sufficient number of adult *Tspan12* KO mice available; we used the available mice to perform the analysis of vascular morphology and density and a sulfo-NHS-biotin permeability assay in adult mutant mice (new Figure 5).

5. It would greatly improve the significance of the study by including additional animal models to investigate the effect of F4L5.13 treatment on retinal vascularization and permeability in oxygen-induced retinopathy and diabetic retinopathy.

This is a good point raised by the reviewer that was also suggested by reviewer #3. We therefore conducted the oxygen-induced retinopathy model and tested the efficacy of F4L5.13 and compared it to anti-VEGF. This required a significant investment of additional resources and time, but we agreed with the reviewer that this was important for the study to demonstrate the efficacy of F4L5.13 in a condition mimicking a disease context other than familial exudative vitreoretinopathy. We believe the manuscript is now significantly stronger describing the effect of F4L5.13 in normalizing the neovascularization seen in the OIR model to levels similar to anti-VEGF treatment.

Referee #3 (Remarks for Author):

In this study, the authors have developed a tetravalent antibody agonist (F4L5.13) of the Norrin/Wnt signalling pathway. F4L5.13 was shown to ablate VEGFA-induced endothelial cell permeability in bEND.3 cells. In *Tspan*^{-/-} mice, used to model impaired retinal vascular development and barrier function, F4L5.13 administration (i.p.) restored post-natal angiogenesis and barrier function in the retina. The effects of the F4L5.13 antibody are impressive, however, some of the in vitro data are questioned (see below) and in other analyses, insufficient information is provided.

We thank the reviewer for the insightful review. We are also excited about the effect of the F4L5.13 antibody.

Specific comments/questions:

1. Blots shown throughout need to be quantified and shown as normalized values relative to the loading control.

This was also raised by other reviewers. We now provide quantification for all the blots.

2. Please show the efficacy of all siRNA knockdowns.

This was included in the supplementary figure of the original manuscript (now Fig EV2). We apologize if we were not clear. We added a note in the text specifically referring to this panel.

3. The number of observations should be clarified. Fields of vision should be shown for number of eyes (one eye/mouse). Overall, the n seems small; for example, in Fig. 4D, n=4. Please clarify that this is 4 mice and not 4 eyes from 2 mice.

We added additional data points and updated the figure legends for Figure 4 and the new Figure 5. For the quantification in the new Figure 4D, five retinas from 4-5 mice per group were analyzed (4 mice in the isotype control group and 5 mice in the other groups). For the new Figure 5 a total of 26 retinas from 13 adult mice were analyzed (6 retinas per group for Figure 5B and 4 retinas per group for Figure 5C.) These experiments are extremely resource intensive in that they required about 10 milligram of both F4L5.13 and control isotype proteins as well as breeding and aging of adult *Tspan12* KO mice from heterozygous intercrosses. All available adult *Tspan12* KO mice were used for this study. To corroborate the findings in Fig5B, we analyzed vascular permeability using sulfo-NHS-biotin labeling and imaging) and consistently found in all experiments that barrier function was partially restored after F4L5.13 treatment.

4. In Fig 1F, the levels of pDVL are considerable in the untreated control and the effect of F4L5.13 at 60 min is similar to the base line, while in the Norrin-treated cells, pDVL levels at 15 min decrease and then perhaps increase again at 60 min (quantification is needed). To make a firm conclusion about kinetics as is done on page 7, first para: "Both Norrin and F4L5.13 led to Dishevelled phosphorylation and bcatenin stabilization with similar kinetics (Fig. 1F).", is not possible. The endothelial cell line used here and in other experiments, b.End.3, is polyoma mT transformed and has a high basal activity. It would be preferable to use primary endothelial cells.

The reviewer is right that these experiments do not allow for a precise description of the kinetics. We now provide quantification for multiple experiments and have tone down the conclusion to simply state that F4L5.13 leads to Dsh3 phosphorylation, an indication that it leads to pathway activity. The high basal Dsh3 phosphorylation in these cells is intriguing. We tried treating the cells with a porcupine inhibitor thinking that the cells may produce autocrine Wnt, but this did not change anything. It may be

that primary cells would exhibit lower levels, but this experiment was merely to provide an orthogonal experiment showing that F4L5.13 led to pathway activation. In addition to β catenin stabilization and target gene activation we believe that the results together provide a strong evidence. As a note, bEnd3 cells respond MUCH better to Norrin than any primary retinal EC we have ever tested.

5. In RNA-seq experiments, what was the effect of F4L5.13 treatment on expression of tight junction proteins (i.e. Claudin-5, ZO-1 and others)? This should be included. Please compare with the effect of Norrin, as normalized expression data.

This was also a question from Reviewer 2. Surprisingly we did not detect changes in gene expression of junction proteins following treatment of NDP or F4L5.13 in Bend3 cells. We validated this using qPCR and detected only non-significant fluctuations. These results indicate that although bEnd3 cells respond to Norrin in a FZD4, TSPAN12 and LRP5 dependent manner, and as a result constitute an interesting cellular system to study signal initiation, the signaling responses and target genes may differ with *in vivo* contexts. Indeed, it is well established that *Cldn5* is regulated by FZD4-bcatenin signaling in mouse retinal endothelial cells (Fig 4E). We added some text to discuss these differences.

6. The overlap of genes induced by F4L5.13 and Norrin was overlapping only to a small extent (77 vs 20) although the authors seem to conclude the majority was overlapping. Please provide a supplemental list of induced genes for the two treatments at the 8 and 24 h time points. Moreover, F4L5.13 treatment appears to show comparable effects with that of Norrin at 8h but at the 24h time points, considerably higher levels of expression of several genes is evident. Was this a dose dependent effect of F4L5.13?

We would like to apologize if we were not clear. What we meant is that the majority of differentially expressed genes in NDP-treated cells were also observed following treatment with F4L5.13. The magnitude of the stimulation is much greater with F4L5.13, which we think led to the identification of additional genes. The F4L5.13 stimulation of gene expression is indeed dose-dependent. We have shown this for *Axin2* (Fig. 1D). We have added a table as requested by the reviewer.

7. The authors make the claim that VEGF breaks down the blood-retinal barrier by disrupting tight junction organization (page 8 and 9). However, VEGF-induced leakage of macromolecules is known to depend on breakdown of adherens junctions (VE-cadherin/b-catenin), not tight junctions. It is possible that VEGF treatment of *in vitro* cultures in which tight junctions may not have become established (due to the transformed nature of the cells or other aspects) may lead to a turnover of *Cldn5* without directly reflecting the quality of the barrier. Importantly, in retinopathies, it is very unlikely that VEGF expression represents the trigger of the barrier breakdown (as is correctly described by the authors on page 9, first para). This does not exclude that anti-VEGF therapy is beneficial by suppressing macromolecular leakage and by improving the perfusion by normalizing the vasculature. Thus, while the data shown in Fig 3 are consistent with literature on that VEGF treatment of endothelial cell lines under certain circumstances can reduce *Cldn5* expression, this has not been validated *in vivo* and the relevance for conclusions on the effect of F4L5.13 on e.g. retinopathy is questionable. It would be best to take the data in Fig. 3B and C out and replace with an *in vivo* model, such as oxygen-induced retinopathy. At the absolute minimum, the authors need to explain how F4L5.13/NDP co-treatment with VEGFA for only 1 hour (too short to allow transcription/translation) preserves CLDN-5 expression the endothelium.

We thank the reviewer for this criticism. We agree that the exact nature of the mechanisms leading to the permeability defects in retinopathies are not completely understood. We however think that there

is strong evidence supporting that VEGF leads to junction disassembly in endothelial cells. Since we believe that the Norrin-FZD4 pathway functions in an alternate pathway to regulate barrier function, we predicted that F4L5.13 would be able to rescue the junction disassembly induced by VEGF. In essence we are using VEGF here just to disassemble the junction and test the ability of F4L5.13 to rescue it through activation of the FZD4 pathway. We however agree with the reviewer that this may not be reflecting the retinopathy context and that using an alternative animal model is justified. The use of the OIR model was also suggested by reviewer 2 and we therefore opted to invest the resources necessary to test F4L5.13 in this model. The results are striking, treatment of mice with F4L5.13 strongly normalize the retina vasculature in the OIR model, similar to anti-VEGF treatment (Fig. 6).

8. According to the literature, Tspan12^{-/-} retinas have a moderate delay in outgrowth of the superficial layer, however, the vertical sprouts and the OPL layer does not form at all. It is not possible from the images in Fig. 4B to estimate the % vascularized area in the Tspan12^{-/-} retinas. This is important as the leakage should be normalized to the vascular area rather than to the weight of the retinas as is done here. Please show the extent of vascularization in the three layers in high and low magnified images.

We now provide a low magnification panel of P20 retinas at extremely high resolution (new Figure 4A). These images were generated by stitching 6 images obtained at 10 x magnification. In addition, we included separate panels showing optical sections of the vasculature in the nerve fiber layer (superficial), inner plexiform layer (intermediate), and outer plexiform layer (deep) at 20x magnification. Furthermore, we show retinal cross sections that demonstrate that vascular development in P20 F4L5.13-treated mice is complete and occurs in all regions of the retina (central and peripheral). F4L5.13-treated Tspan12 KO retinas at P20 are difficult to distinguish from wildtype retinas, perhaps with the exception that PLVAP staining intensity is a little higher than in wildtype retinas (but clearly lower than in KO retinas). In contrast to the full restoration of vascular development after administration of multiple doses from P6 onward, we observed only partial restoration of vascular morphogenesis by P20 when a single dose of F4L5.13 was administered at P9 (not shown.) F4L5.13 treatment in adult Tspan12 KO mice did not cause a normalization of the aberrant mutant vasculature or increase vascular density. F4L5.13 treatment is therefore not sufficient to activate angiogenesis in the quiescent adult vasculature. We included high resolution 4x images of the entire adult retina as well as projections of the aberrant mutant vasculature at 20x magnification. Since the mutant vasculature does not respect layer boundaries of the superficial and intermediate plexuses, we did not attempt to separate these layers by optical sectioning and therefore show projections of all blood vessels in the mutant retina (with and without treatment) at 20x (new Figure 5). Please see also our response to reviewer 1 point 7, and reviewer 2 point 2.

9. The Tspan12^{-/-} is an interesting model to demonstrate the effect of F4L5.13, however, it is a genetic model rather than a disease model. It would be both important and very valuable if the authors complement their study with an in vivo retinopathy model.

We agree. And as mentioned above we have performed the recommended experiments. We would like to thank the reviewer for this suggestion that significantly strengthen the manuscript.

Minor

10. The format of author names in the reference section is inconsistent. In some instances they include first author's name followed by "et al."

We have reviewed the references.

28th Apr 2021

Dear Stephane,

Thank you for the submission of your revised manuscript to EMBO Molecular Medicine, and please accept my apologies for the delay in getting back to you, which is due to the fact that two referees needed more time to provide their reports and that we currently experience a high load of submissions to our editorial office.

We have now received the enclosed reports from the referees. As you will see, they are supportive of publication, and I am therefore pleased to inform you that we will be able to accept your manuscript, once the following minor points will be addressed:

1) Please address the points from referee #3.

2) Main manuscript text:

- Please answer/correct the changes suggested by our data editors in the main manuscript file (in track changes mode). This file will be sent to you in the next couple of days. Please use this file for any further modification.

- Please remove the highlights in the text.

- Material and methods:

- o Cells: please indicate the origin of the cells, and whether they were tested for mycoplasma contamination

- o Mice: please indicate the origin of the mice and their housing and husbandry conditions (in the checklist as well).

- Thank you for providing a Data Availability section. Please place it after the Material and Methods and provide an URL. Please note that the data has to be public before publication of the manuscript.

- Author contributions: Somasekar Seshagiri is missing from this section.

- Please confirm that there is a total of 3 co-corresponding authors for this manuscript.

3) Figures and tables:

- Statistics: Please indicate in the figures or in the legends the exact n= and exact p= values along with the statistical test used. You may provide these values as a supplemental table in an Appendix file.

- Please add their legends to the 2 EV tables excel files.

4) Thank you for providing The Paper Explained. I added minor modifications to shorten the section, please let me know if you agree with the following:

Problem

Retinal homeostasis requires an intact blood retina barrier (BRB) and BRB dysfunction is associated with retinal diseases. In addition, retinal homeostasis requires a proper balance of angiogenesis vs. vascular quiescence. In the retina, the secreted ligand Norrin (NDP) binds FZD4 and the co-receptors LRP5 and TSPAN12 and leads to β catenin- dependent development of the retinal vascular and BRB formation. Mutations in genes important for this pathway lead to rare congenital eye diseases, such as Norrie disease and FEVR, in which the retinal vasculature and BRB is

disrupted, causing impaired vision or blindness. More prevalent diseases such as diabetic retinopathy and macular degeneration are also characterized by neovascularization and blood retina barrier defects. New therapeutic approaches are needed that actively restore BRB function and normalize aberrant retinal vasculatures.

Results

We have engineered F4L5.13, a tetravalent antibody, designed to induce FZD4 and LRP5 proximity in order to activate downstream β catenin signaling. We demonstrated that it is selective for FZD4 and that it efficiently activates β -catenin signaling in mouse brain endothelial cells (bEnd.3 cells) to levels surpassing NDP. F4L5.13 mimics the activity of NDP by activating FZD4 and LRP5 but without requiring TSPAN12. F4L5.13 promotes barrier function in endothelial cells in vitro. In the Tspan12^{-/-} mouse model that exhibits several features of FEVR, systemic injection of F4L5.13 rescued the blood vessel morphogenesis and BRB defects. Finally, when tested in the oxygen-induced retinopathy model, which exhibits hallmarks of neovascular diseases such as retinopathy of prematurity, wet macular degeneration and diabetic retinopathy, the FZD4:LRP5 antibody agonist was found to normalize the observed pathological neovascularization.

Impact

F4L5.13 is a novel synthetic tetravalent antibody that allows the precise (FZD4 and LRP5 specific) and potent activation of β -catenin signaling in vitro and in vivo. This antibody therefore represents a first-in-class antibody to treat ocular diseases as well as stroke and neurological disorders where endothelial cell barrier function is impaired.

5) Thank you for providing a nice synopsis picture. Please also provide a text to further enhance discoverability. It should include a short stand first (maximum of 300 characters, including space) as well as 2-5 one-sentences bullet points that summarizes the paper. Please write the bullet points to summarize the key NEW findings. They should be designed to be complementary to the abstract - i.e. not repeat the same text. We encourage inclusion of key acronyms and quantitative information (maximum of 30 words / bullet point).

6) As part of the EMBO Publications transparent editorial process initiative (see our Editorial at <http://embomolmed.embopress.org/content/2/9/329>), EMBO Molecular Medicine will publish online a Review Process File (RPF) to accompany accepted manuscripts.

This file will be published in conjunction with your paper and will include the anonymous referee reports, your point-by-point response and all pertinent correspondence relating to the manuscript. Let us know whether you agree with the publication of the RPF.

I look forward to receiving your revised manuscript.

With kind regards,

Lise

Lise Roth, PhD
Editor
EMBO Molecular Medicine

To submit your manuscript, please follow this link:

<https://embomolmed.msubmit.net/cgi-bin/main.plex>

***Additional important information regarding Figures**

Photos 400-800 DPI

*Additional important information regarding figures and illustrations can be found at

The system will prompt you to fill in your funding and payment information. This will allow Wiley to send you a quote for the article processing charge (APC) in case of acceptance. This quote takes into account any reduction or fee waivers that you may be eligible for. Authors do not need to pay any fees before their manuscript is accepted and transferred to our publisher.

***** Reviewer's comments *****

Referee #1 (Comments on Novelty/Model System for Author):

All the molecular mechanism has been already published and well known. However, if the journal scope fit with the claim of this study (implication to clinical utility), this paper is acceptable.

Referee #1 (Remarks for Author):

In this revised paper, authors have sufficiently addressed all of my criticisms and strengthened the data. Now the paper is acceptable.

Referee #2 (Remarks for Author):

The authors are very responsive to reviewers' comments. All my concerns have been addressed adequately. The revised manuscript is substantially improved in particular with the addition of a disease model of oxygen-induced retinopathy.

Referee #3 (Comments on Novelty/Model System for Author):

The reagent described here, the Norrin/Wnt surrogate antibody F4L5.13, is convincingly shown to enforce the blood brain barrier and has potential medical applications in certain eye diseases characterised by excessive leakage. The technical quality is OK but not striking. With the added oxygen-driven retinopathy model, it's better now in the revised version, still a substantial amount of data is generated using a transformed endothelial cell line, bEnd.

Referee #3 (Remarks for Author):

The revision has improved the quality of the presentation. In particular, the addition of the oxygen-induced retinopathy model is interesting. Please provide the weight of the pups at P17 to inform on whether the treatment has affected the general well-being of the pups. A lower weight may explain the reduced neoangiogenesis. To give pup weights is standard in the field, and if not available, this should be stated as a limitation.

In the image of treated retinas in Fig. 6D, pups receiving the high dose of F4L5.13 appear to have markedly dilated retinal veins. Was this a common phenomenon? Please comment and provide quantification. Alternatively if not representative then replace the image.

When commenting on the limitations of the bEnd cell line, i.e., in not agreeing with in vivo models, please add that bEnd is a polyoma mT transformed line. This is important information.

**** Reviewer's comments ****

Referee #1 (Comments on Novelty/Model System for Author):

All the molecular mechanism has been already published and well known. However, if the journal scope fit with the claim of this study (implication to clinical utility), this paper is acceptable.

Referee #1 (Remarks for Author):

In this revised paper, authors have sufficiently addressed all of my criticisms and strengthened the data. Now the paper is acceptable.

We thank the reviewer for providing helpful comments that have altogether improved the quality of the study.

Referee #2 (Remarks for Author):

The authors are very responsive to reviewers' comments. All my concerns have been addressed adequately. The revised manuscript is substantially improved in particular with the addition of a disease model of oxygen-induced retinopathy.

We thank the reviewer for providing helpful comments that have altogether improved the quality of the study.

Referee #3 (Comments on Novelty/Model System for Author):

The reagent described here, the Norrin/Wnt surrogate antibody F4L5.13, is convincingly shown to enforce the blood brain barrier and has potential medical applications in certain eye diseases characterised by excessive leakage. The technical quality is OK but not striking. With the added oxygen-driven retinopathy model, it's better now in the revised version, still a substantial amount of data is generated using a transformed endothelial cell line, bEnd.

We thank the reviewer for the insightful review.

Referee #3 (Remarks for Author):

The revision has improved the quality of the presentation. In particular, the addition of the oxygen-induced retinopathy model is interesting. Please provide the weight of the pups at P17 to inform on whether the treatment has affected the general well-being of the pups. A lower weight may explain the reduced neoangiogenesis. To give pup weights is standard in the field, and if not available, this should be stated as a limitation.

We did not weigh the animals prior to sacrifice. From our anecdotal observations, these animals were of normal weight, physiology, and behavior at time of sacrifice. Individuals highly experienced in mouse handling ran these experiments and thus abnormalities in pup health would be easily noticeable and noted. Prior to sacrifice, there wasn't any attrition that could be attributed to a specific treatment group. Also, as the antibodies were administered locally via

intravitreal injection, we wouldn't expect to see systemic effects like weight loss indicating adequate drug tolerance. As suggested by the reviewer we have added this information in the methods.

In the image of treated retinas in Fig. 6D, pups receiving the high dose of F4L5.13 appear to have markedly dilated retinal veins. Was this a common phenomenon? Please comment and provide quantification. Alternatively if not representative then replace the image.

We have reviewed all retinal flatmount images from this experiment. The dilation of the retinal veins noticed by the reviewer was not a common observation in any treatment group. Therefore, as suggested by the reviewer we have replaced the relevant panel in Fig. 6D with a more representative image.

When commenting on the limitations of the bEnd cell line, i.e., in not agreeing with in vivo models, please add that bEnd is a polyoma mT transformed line. This is important information.

We have added this information in the discussion.

17th May 2021

Dear Stephane,

Thank you for sending the revised files. I have looked at everything, and all is fine. I am therefore very pleased to accept your manuscript for publication in EMBO Molecular Medicine!

Please note that I slightly modified the synopsis to fit our style (use of passive voice):

'This study reports a FZD4:LRP5 antibody agonist (F4L5.13) that activates β catenin signaling in endothelial cells. F4L5.13 shows efficacy in animal models by normalizing defective retinal angiogenesis and barrier function, providing a novel therapeutic strategy for eye diseases.

- β catenin signaling was activated by F4L5.13, which functions as a Norrin surrogate in endothelial cells.
- Endothelial barrier function was promoted, and VEGF-induced endothelial permeability was blocked by F4L5.13.
- Retinal barrier function was restored by F4L5.13 in a Tspan12^{-/-} mice.
- Pathological neovascularization was reduced by F4L5.13 in an OIR model.'

Please contact us immediately if you do not agree.

Your manuscript will be sent to our publisher to be included in the next available issue of EMBO Molecular Medicine.

Congratulations on a nice study!

With my best wishes,

Lise

Lise Roth, Ph.D
Editor
EMBO Molecular Medicine

Follow us on Twitter @EmboMolMed
Sign up for eTOCs at embopress.org/alertsfeeds

YOU MUST COMPLETE ALL CELLS WITH A PINK BACKGROUND ↓
PLEASE NOTE THAT THIS CHECKLIST WILL BE PUBLISHED ALONGSIDE YOUR PAPER

Corresponding Author Name: Stephane Angers
Journal Submitted to: EMBO Molecular Medicine
Manuscript Number: EMM-2021-13977